# NeuroSED: Learning Subgraph Similarity via Graph Neural Networks

## Abstract

Subgraph similarity search is a fundamental operator in graph analysis. In this framework, given a query graph and a graph database, the goal is to identify subgraphs of the database graphs that are structurally similar to the query. *Subgraph edit distance* (SED) is one of the most expressive measures of subgraph similarity. In this work, we study the problem of *learning* SED from a training set of graph pairs and their SED values. Towards that end, we design a novel *siamese* graph neural network called NeuroSED, which learns an embedding space with a rich structure reminiscent of SED. With the help of a specially crafted inductive bias, NeuroSED not only enables high accuracy but also ensures that the predicted SED, like true SED, satisfies the *triangle inequality*. The design is generic enough to also model *graph edit distance* (GED), while ensuring that the predicted GED space is *metric*, like the true GED space. Extensive experiments on real graph datasets, for both SED and GED, establish that NeuroSED achieves $\approx 2$ times lower RMSE than the state of the art and is $\approx 18$ times faster than the fastest baseline. Further, owing to its pair-independent embeddings and theoretical properties, NeuroSED allows up to 3 orders of magnitude faster graph/subgraph retrieval.

## 1 Introduction and Related work

Graphs are used to model data in a wide variety of domains. Examples include chemical compounds (Sankar et al., 2017), protein-protein interaction networks (PPI) (Alon et al., 2008), knowledge graphs (Ebsch et al., 2020), and social networks (Kempe et al., 2003). A distance function on any dataset, including graphs, is a fundamental operator. Among several distance measures on graphs, *edit distance* is one of the most powerful and popular mechanisms (Liang & Zhao, 2017; Zhao et al., 2013; Bougleux et al., 2017; Daller et al., 2018; Fankhauser et al., 2011; Riesen & Bunke, 2009; Zeng et al., 2009; He & Singh, 2006). Edit distance can be posed in two forms: *graph edit distance* (GED) and *subgraph edit distance* (SED). Given two graphs $\mathcal{G}_1$ and $\mathcal{G}_2$, $\mathrm{GED}(\mathcal{G}_1, \mathcal{G}_2)$ returns the minimum cost of *edits* needed to convert $\mathcal{G}_1$ to $\mathcal{G}_2$. i.e., for $\mathcal{G}_1$ to become *isomorphic* to $\mathcal{G}_2$. An edit can be the addition or deletion of edges and nodes, or replacement of edge or node labels, with an associated cost. In $\mathrm{SED}(\mathcal{G}_1, \mathcal{G}_2)$, the goal is to identify the minimum cost of edits so that $\mathcal{G}_1$ is a subgraph (*subgraph isomorphic*) of $\mathcal{G}_2$. For examples, see Fig. 5a in the appendix.

While the applications of GED and SED are beyond doubt, their applicability is constrained by their computation costs. Specifically, both GED and SED are NP-complete (Zeng et al., 2009; Neuhaus et al., 2006; He & Singh, 2006). To mitigate this computational bottleneck, several heuristics (Bougleux et al., 2017; Daller et al., 2018; Fankhauser et al., 2011; Riesen & Bunke, 2009; Neuhaus et al., 2006) and index structures (He & Singh, 2006; Zeng et al., 2009; Liang & Zhao, 2017; Zhao et al., 2013) have been proposed. Recently, graph neural networks have been shown to be effective in learning and predicting GED (Bai et al., 2019; Wang et al., 2021; Li et al., 2019; Bai et al., 2020; Zhang et al., 2021; Xiu et al., 2020) and subgraph isomorphism (Rex et al., 2020). The basic goal in all these algorithms is to learn a neural model from a training set of graph pairs and their distances, such that, at inference time, given an unseen graph pair, we are able to predict its distance accurately. Wang et al. (2020) designed an algorithm to characterize graphs through sampling random walks and then embedding them in a feature space. While Wang et al. (2020) has not been studied in the context of GED or SED, the authors have shown that they can detect graph isomorphism. While this progress in the area of graph querying is undoubtedly impressive, there is scope to do more.

- **Modeling SED:** Existing neural approaches to learning GED cannot easily be adapted to learn SED. While GED is symmetric, SED is not. Most neural architectures for GED have the assumption of symmetry at its core and hence modeling SED is non-trivial. In App. A, we provide a detailed analysis of the limitations of prior architectures in failing to model SED.

- **Exponential Search Space:** Computing $\text{SED}(\mathcal{G}_1, \mathcal{G}_2)$ conceptually requires us to compare the query graph $\mathcal{G}_1$ with the exponentially many subgraphs of the target graph $\mathcal{G}_2$. Therefore, it is imperative that the model has an efficient and effective mechanism to prune the search space without compromising on the prediction accuracy. We note that while several index structures and heuristics exist for GED (He & Singh, 2006; Zeng et al., 2009; Bai et al., 2019; Wang et al., 2021), none exist for SED. Thus, scalability of SED on large graphs remains an unsolved problem.
- **Preservation of theoretical properties:** GED is a *metric* distance function. While SED is not metric due to being asymmetric, it satisfies the *triangle inequality*, *non-negativity*, and *subgraph-identity*. Several higher-order tasks such as clustering and indexing rely on such properties (Hadi, 1991; Dohnal et al., 2003; Hjaltason & Samet, 2003; Samet, 2005; Uhlmann, 1991; Yianilos, 1993; Zezula et al., 2006; Brin, 1995; Chávez et al., 2001). Existing neural approaches do not preserve these properties, which limits their usability for these higher order tasks.
- **Pair-independent computation:** There is little scope for pre-computation in existing approaches (either neural or non-neural), as the major computations are pair-dependent, i.e., both $\mathcal{G}_1$ and $\mathcal{G}_2$ need to be known. In a typical graph querying framework, the graph database is known apriori; only the query graph is provided at runtime. If we can generate *pair-independent* embeddings, and make efficient predictions directly in the embedding space, then retrieval can be sped up tremendously by pre-computing and indexing the embeddings of the database graphs beforehand.
- **Learning with low-volume data:** Generating a large amount of high quality training data is computationally prohibitive for SED/GED. So, an effective model should encode powerful priors for SED/GED prediction, to enable generalization even with scarce training data.

We address these challenges by a novel architecture called NEUROSED. Our key contributions are:

- **Novel formulation:** We formulate the problem of learning subgraph edit distance (SED) for graphs. Since SED is more general than subgraph isomorphism and GED, the proposed theory extends to prediction of GED, graph isomorphism and subgraph isomorphism.
- **Neural architecture:** NEUROSED utilizes a *siamese graph isomorphism network* (Xu et al., 2018a) to embed graphs in a *pair-independent* fashion to an embedding space over which a simple function can predict the SED (or GED). The same embedding model is used for both $\mathcal{G}_1$ and $\mathcal{G}_2$, which captures the prior that similar topological properties need to be considered for both graphs. The carefully crafted prediction function serves as another *inductive bias* for the model, which, in addition to enabling high generalization accuracy, preserves the key properties of SED/GED, which is a major contribution of our work over existing neural approaches.
- **Indexable embeddings:** The prediction function satisfies the triangle inequality over the embedding space for both SED and GED. This allows utilization of the rich literature on index structures (Elkan, 2003; Kwedlo & Czochanski, 2019; Ciaccia et al., 1997) to boost efficiency.
- **Accurate, Fast and Scalable:** Extensive experiments on real graph datasets containing up to a million nodes establish that NEUROSED is more accurate in both GED and SED when compared to the state of the art and is more than 3 orders of magnitude faster in range and $k$-NN queries.

## 2 PRELIMINARIES AND PROBLEM FORMULATION

We denote a labeled undirected graph as $\mathcal{G} = (\mathcal{V}, \mathcal{E}, \mathcal{L})$ where $\mathcal{V}$ is the node set, $\mathcal{E}$ is the edge set and $\mathcal{L} : \mathcal{V} \cup \mathcal{E} \to \Sigma$ is the labeling function over nodes and edges. $\Sigma$ is the universe of all labels and contains a special empty label $\epsilon$. $\mathcal{L}(v)$ and $\mathcal{L}(e)$ denote the labels of node $v$ and edge $e$ respectively. $\mathcal{G}_1 \subseteq \mathcal{G}_2$ denotes that $\mathcal{G}_1$ is a *subgraph* of $\mathcal{G}_2$. All notations used in our work are summarized in Table 6 in the appendix. For the definitions of *subgraph and graph isomorphism*, refer to App. B in the supplementary. The computation of GED relies on a *graph mapping*.

**Definition 1 (Graph Mapping)** *Given two graphs $\mathcal{G}_1$ and $\mathcal{G}_2$, let $\tilde{\mathcal{G}}_1 = (\tilde{\mathcal{V}}_1, \tilde{\mathcal{E}}_1, \tilde{\mathcal{L}}_1)$ and $\tilde{\mathcal{G}}_2 = (\tilde{\mathcal{V}}_2, \tilde{\mathcal{E}}_2, \tilde{\mathcal{L}}_2)$ be obtained by adding dummy nodes and edges (labeled with $\epsilon$) to $\mathcal{G}_1$ and $\mathcal{G}_2$ respectively, such that $|\mathcal{V}_1| = |\mathcal{V}_2|$ and $|\mathcal{E}_1| = |\mathcal{E}_2|$. A node mapping between $\mathcal{G}_1$ and $\mathcal{G}_2$ is a bijection $\pi : \tilde{\mathcal{G}}_1 \to \tilde{\mathcal{G}}_2$ where (i) $\forall v \in \tilde{\mathcal{V}}_1, \pi(v) \in \tilde{\mathcal{V}}_2$ and at least one of $v$ and $\pi(v)$ is not a dummy; (ii) $\forall e = (v_1, v_2) \in \tilde{\mathcal{E}}_1, \pi(e) = (\pi(v_1), (\pi(v_2))) \in \tilde{\mathcal{E}}_2$ and at least one of $e$ and $\pi(e)$ is not a dummy.*

**Example 1** *Fig. 5b shows a graph mapping. Edge mappings can be trivially inferred, so are omitted.*

**Definition 2 (Graph Edit Distance (GED) under mapping $\pi$)** GED *between $\mathcal{G}_1$ and $\mathcal{G}_2$ under $\pi$ is*

$$\text{GED}_\pi(\mathcal{G}_1, \mathcal{G}_2) = \sum_{v \in \tilde{V}_1} d(\mathcal{L}(v), \mathcal{L}(\pi(v))) + \sum_{e \in \tilde{E}_1} d(\mathcal{L}(e), \mathcal{L}(\pi(e))) \quad (1)$$

*where $d : \Sigma \times \Sigma \to \mathbb{R}_0^+$ is a distance function over the label set.*

$d(\ell_1, \ell_2)$ models an *insertion* if $\ell_1 = \epsilon$, *deletion* if $\ell_2 = \epsilon$ and *replacement* if $\ell_1 \neq \ell_2$ and neither $\ell_1$ nor $\ell_2$ is a dummy. We assume $d$ to be a binary function, where $d(\ell_1, \ell_2) = 1$ if $\ell_1 \neq \ell_2$, otherwise, $0$. Our framework easily extends to more general distance functions (details in App. D).

**Example 2** GED *for the shown mapping in Fig. 5b is 3. The red mappings incur a cost of 1.*

**Definition 3 (Graph Edit Distance (GED))** GED *is the minimum distance under all mappings.*
$$\text{GED}(\mathcal{G}_1, \mathcal{G}_2) = \min_{\forall \pi \in \Phi(\mathcal{G}_1, \mathcal{G}_2)} \text{GED}_\pi(\mathcal{G}_1, \mathcal{G}_2) \tag{2}$$

*where $\Phi(\mathcal{G}_1, \mathcal{G}_2)$ denotes the set of all possible node maps from $\mathcal{G}_1$ to $\mathcal{G}_2$.*

**Definition 4 (Subgraph Edit Distance (SED))** SED *is the minimum GED over all subgraphs of $\mathcal{G}_2$.*
$$\text{SED}(\mathcal{G}_1, \mathcal{G}_2) = \min_{\mathcal{S} \subseteq \mathcal{G}_2} \text{GED}(\mathcal{G}_1, \mathcal{S}) \tag{3}$$

**Example 3** *Revisiting Fig. 5b, $\text{GED}(\mathcal{G}_1, \mathcal{G}_2) = 3$. The shown mapping incurs the minimum cost. $\text{SED}(\mathcal{G}_1, \mathcal{G}_2) = 1$. Refer to Fig. 5a for more examples.*

**Problem 1 (Learning SED)** *Given a training set of tuples of the form $\langle \mathcal{G}_1, \mathcal{G}_2, \text{SED}(\mathcal{G}_1, \mathcal{G}_2) \rangle$, learn a neural model to predict $\text{SED}(\mathcal{Q}_1, \mathcal{Q}_2)$ on unseen graphs $\mathcal{Q}_1$ and $\mathcal{Q}_2$.*

**Problem 2 (Learning GED)** *Given a training set of tuples of the form $\langle \mathcal{G}_1, \mathcal{G}_2, \text{GED}(\mathcal{G}_1, \mathcal{G}_2) \rangle$, learn a neural model to predict $\text{GED}(\mathcal{Q}_1, \mathcal{Q}_2)$ on unseen graphs $\mathcal{Q}_1$ and $\mathcal{Q}_2$.*

### 2.1 PROPERTIES OF SED AND GED

**Observation 1** *(i)* $\text{GED}(\mathcal{G}_1, \mathcal{G}_2) \geq 0$, *(ii)* $\text{SED}(\mathcal{G}_1, \mathcal{G}_2) \geq 0$.

**Observation 2** *(i)* $\text{GED}(\mathcal{G}_1, \mathcal{G}_2) = 0$ *iff $\mathcal{G}_1$ is isomorphic to $\mathcal{G}_2$, (ii)* $\text{SED}(\mathcal{G}_1, \mathcal{G}_2) = 0$ *iff $\mathcal{G}_1$ is subgraph isomorphic to $\mathcal{G}_2$.*

**Observation 3** GED *is a metric if the distance function $d$ over label set $\Sigma$ is metric (He & Singh, 2006). $d$ as defined in § 2 is metric (He & Singh, 2006). Hence, GED from § 2 is metric.*

We next prove that SED satisfies the triangle inequality in Theorem 1.

**Theorem 1** $\text{SED}(\mathcal{G}_1, \mathcal{G}_3) \leq \text{SED}(\mathcal{G}_1, \mathcal{G}_2) + \text{SED}(\mathcal{G}_2, \mathcal{G}_3)$.

PROOF: See App. D.1 for details.

## 3 NEUROSED

Fig. 1 presents the architecture of NEUROSED. The input to our learning framework is a pair of graphs $\mathcal{G}_\mathcal{Q}$ (query), $\mathcal{G}_\mathcal{T}$ (target) along with the supervision data $\text{SED}(\mathcal{G}_Q, \mathcal{G}_T)$. Our objective is to train a model that can predict SED on unseen query and target graphs. The design of our model must be cognizant of the fact that computing SED is NP-hard and high quality training data is scarce. Thus, we use a *Siamese* architecture in which the weight sharing between the embedding models boosts learnability and generalization from low-volume data by imposing a strong prior that the same topological features must be extracted for both graphs.
**Siamese Networks:** In these models, there are *two* networks with *shared* parameters applied to two inputs independently to compute representations. These representations are then passed through another module to compute a similarity score.

### 3.1 SIAMESE GRAPH NEURAL NETWORK

As depicted in Fig. 1a, we use a siamese graph neural network (GNN) with shared parameters to embed both $\mathcal{G}_\mathcal{Q}$ and $\mathcal{G}_\mathcal{T}$. Fig. 1b focuses on the GNN component of NEUROSED. We next discuss each of its individual components.
**Pre-MLP:** The primary task of the Pre-MLP is to learn representations for the node labels (or features). Towards that end, let $\mathbf{x}_v$ denote the initial feature set of node $v$. The MLP learns a hidden representation $\boldsymbol{\mu}_v^\mathcal{G} = \text{MLP}(\mathbf{x}_v)$. In our implementation, $\mathbf{x}_v$ is a one-hot encoding of the categorical node labels. We do not explicitly model edge labels in our experiments. NEUROSED can easily be extended to edge labels by strategies such as using GINE (Hu et al., 2019) layers instead of GIN layers (see App. C).

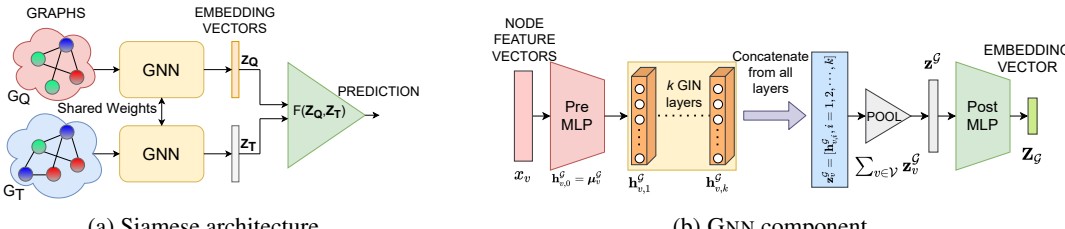

(a) Siamese architecture               (b) GNN component

Figure 1: **The architecture of NEUROSED.**

**Graph Isomorphism Network (GIN):** GIN (Xu et al., 2018b) consumes the information from the Pre-MLP to learn hidden representations that encode both the graph structure as well as the node feature information. GIN is as powerful as the *Weisfeiler-Lehman (WL) graph isomorphism test* (Leman & Weisfeiler, 1968) in distinguishing graph structures. Since our goal is to accurately characterize graph topology and learn similarity, GIN emerges as the natural choice. GIN develops its expressive power by using an *injective* aggregation function. Specifically, in the initial layer, each node $v$ in graph $\mathcal{G}$ is characterized by the representation learned by the MLP, i.e., $\mathbf{h}_{v,0}^{\mathcal{G}} = \boldsymbol{\mu}_v^{\mathcal{G}}$. Subsequently, in each hidden layer $i$, we learn an embedding through the following transformation.

$$\mathbf{h}_{v,i}^{\mathcal{G}} = \text{MLP}\left((1 + \epsilon^i) \cdot \mathbf{h}_{v,i-1}^{\mathcal{G}} + \sum_{u \in \mathcal{N}_{\mathcal{G}}(v)} \mathbf{h}_{u,i-1}^{\mathcal{G}}\right) \tag{4}$$

Here, $\epsilon^i$ is a layer-specific learnable parameter, $\mathcal{N}_{\mathcal{G}}(v)$ is one-hop neighbourhood of the node $v$, and $\mathbf{h}_{v,0}^{\mathcal{G}} = \boldsymbol{\mu}_v^{\mathcal{G}}$. The $k$-th layer embedding is $\mathbf{h}_{v,k}^{\mathcal{G}}$, where $k$ is final hidden layer.

**Concatenation, Pool and Post-MLP:** Intuitively, $\mathbf{h}_{v,i}^{\mathcal{G}}$ captures a feature-space representation of the $i$-hop neighborhood of $v$. Typically, GNNs operate on node or edge level predictive tasks, such as node classification or link prediction, and hence, the node representations are passed through an MLP for the final prediction task. In our problem, we need to capture a graph level representation. Furthermore, the representation should be rich enough to also capture the various subgraphs within the input graph so that SED can be predicted accurately. To fulfil these requirements, we first *concatenate* the representation of a node across *all* hidden layers, i.e., the final node embedding is $\mathbf{z}_v^{\mathcal{G}} = \text{CONCAT}\left(\mathbf{h}_{v,i}^{\mathcal{G}}, \forall i \in \{1, 2, \cdots, k\}\right)$. This allows us to capture a multi-granular view of the subgraphs centered on $v$ at different radii in the range $[1, k]$. Next, to construct the graph-level representation, we perform a sum-pool, which adds the node representations to give a single vector. This information is then fed to the Post-MLP to enable post-processing. Mathematically:

$$\mathbf{Z}_{\mathcal{G}} = \text{MLP}(\mathbf{z}^{\mathcal{G}}) = \text{MLP}\left(\sum_{v \in \mathcal{V}} \mathbf{z}_v^{\mathcal{G}}\right) \tag{5}$$

**SED Prediction:** The final task is to predict the SED as a function of query graph embedding $\mathbf{Z}_{\mathcal{G}_{\mathcal{Q}}}$ and target graph embedding $\mathbf{Z}_{\mathcal{G}_{\mathcal{T}}}$. The natural choice would be to feed these embeddings into another MLP to learn $\text{SED}(\mathbf{Z}_{\mathcal{G}_{\mathcal{Q}}}, \mathbf{Z}_{\mathcal{G}_{\mathcal{T}}})$. This MLP can then be trained jointly with the graph embedding model in an *end-to-end* fashion. However, an MLP prediction does not have any theoretical guarantees.

To ensure preservation of original space properties, we introduce an *inductive bias* in the form of a simple fixed computation function $\mathcal{F}(\mathbf{Z}_{\mathcal{G}_{\mathcal{Q}}}, \mathbf{Z}_{\mathcal{G}_{\mathcal{T}}})$ to predict the SED. By training the model to produce embeddings $\mathbf{Z}_{\mathcal{G}_{\mathcal{Q}}}$ and $\mathbf{Z}_{\mathcal{G}_{\mathcal{T}}}$ such that $\mathcal{F}(\mathbf{Z}_{\mathcal{G}_{\mathcal{Q}}}, \mathbf{Z}_{\mathcal{G}_{\mathcal{T}}}) \approx \text{SED}(\mathbf{Z}_{\mathcal{G}_{\mathcal{Q}}}, \mathbf{Z}_{\mathcal{G}_{\mathcal{T}}})$, we enforce a rich structure on the embedding space. We show that these embeddings satisfy many key properties of the SED (and GED) function that existing neural algorithms fail to do (Bai et al., 2019; Wang et al., 2021; Bai et al., 2020; Zhang et al., 2021). $\mathcal{F}$ is defined as follows:

$$\mathcal{F}(\mathbf{Z}_{\mathcal{G}_{\mathcal{Q}}}, \mathbf{Z}_{\mathcal{G}_{\mathcal{T}}}) = \|ReLU(\mathbf{Z}_{\mathcal{G}_{\mathcal{Q}}} - \mathbf{Z}_{\mathcal{G}_{\mathcal{T}}})\|_2 = \|\max\{0, \mathbf{Z}_{\mathcal{G}_{\mathcal{Q}}} - \mathbf{Z}_{\mathcal{G}_{\mathcal{T}}}\}\|_2 \tag{6}$$

Intuitively, for those co-ordinates where the value of $\mathbf{Z}_{\mathcal{G}_{\mathcal{Q}}}$ is greater than $\mathbf{Z}_{\mathcal{G}_{\mathcal{T}}}$; a distance penalty is accounted by $\mathcal{F}$ in terms of how much those values differ, otherwise $\mathcal{F}$ considers 0. This follows the intuition that the SED accounts for those features of $\mathcal{G}_{\mathcal{Q}}$ that are not in $\mathcal{G}_{\mathcal{T}}$. Moreover, consistent with SED, the additional features in $\mathcal{G}_{\mathcal{T}}$ that are not in $\mathcal{G}_{\mathcal{Q}}$, do not incur any cost. Finally, the parameters of the entire model are learned by minimizing the mean squared error[1] (here $\mathbb{T}$ is the training set).

$$\mathscr{L} = \frac{1}{|\mathbb{T}|} \sum_{\forall \langle \mathcal{G}_{\mathcal{Q}}, \mathcal{G}_{\mathcal{T}} \rangle \in \mathbb{T}} \left(\mathcal{F}(\mathbf{Z}_{\mathcal{G}_{\mathcal{Q}}}, \mathbf{Z}_{\mathcal{G}_{\mathcal{T}}}) - \text{SED}(\mathcal{G}_{\mathcal{Q}}, \mathcal{G}_{\mathcal{T}})\right)^2 \tag{7}$$

---

[1]Since SED computation is NP-hard, in App. G, we discuss a loss function based on SED lower and upper bounds, which are faster to compute.

**Adaptation for GED (NEUROSED$_\mathcal{G}$):** The proposed architecture naturally extends to GED (denoted by NEUROSED$_\mathcal{G}$) with a simple modification of the computation function. Specifically, instead of Eq. 6, we use

$$\mathcal{F}_g(\mathbf{Z}_{\mathcal{G}_\mathcal{Q}}, \mathbf{Z}_{\mathcal{G}_\mathcal{T}}) = \|\mathbf{Z}_{\mathcal{G}_\mathcal{Q}} - \mathbf{Z}_{\mathcal{G}_\mathcal{T}}\|_2 \qquad (8)$$

## 3.2 Theoretical Characterization

**Lemma 1** *The following properties hold on predicted* SED $\mathcal{F}(\mathbf{Z}_{\mathcal{G}_\mathcal{Q}}, \mathbf{Z}_{\mathcal{G}_\mathcal{T}})$:

1. $\mathcal{F}(\mathbf{Z}_{\mathcal{G}_\mathcal{Q}}, \mathbf{Z}_{\mathcal{G}_\mathcal{T}}) \geq 0$
2. $\mathcal{F}(\mathbf{Z}_{\mathcal{G}_\mathcal{Q}}, \mathbf{Z}_{\mathcal{G}_\mathcal{T}}) = 0 \iff \mathbf{Z}_{\mathcal{G}_\mathcal{Q}} \leq \mathbf{Z}_{\mathcal{G}_\mathcal{T}}$
3. $\mathcal{F}(\mathbf{Z}_{\mathcal{G}_\mathcal{Q}}, \mathbf{Z}_{\mathcal{G}_\mathcal{T}}) \leq \mathcal{F}(\mathbf{Z}_{\mathcal{G}_\mathcal{Q}}, \mathbf{Z}_{\mathcal{G}_{\mathcal{T}'}}) + \mathcal{F}(\mathbf{Z}_{\mathcal{G}_{\mathcal{T}'}}, \mathbf{Z}_{\mathcal{G}_\mathcal{T}})$

PROOF. Properties (1) and (2) follow from the definition of $\mathcal{F}$ itself. Property (3) follows from the fact that we take the $L_2$ norm (holds for any monotonic norm, details are available in App. D). □

We see that $\mathcal{F}$ satisfies the analogue of properties of SED (§ 2.1). Conditions 1, 2, and 3 captures non-negativity, the subgraph relation (Rex et al., 2020), and triangle inequality respectively.

Furthermore, we show in App. E a complete constructive characterization for any $\mathcal{F}$ which can model SED in the embedding space, while being translation-invariant and non-negative homogeneous. Specifically, we show that the general form for $\mathcal{F}$ involves solving a constrained optimization problem (intractable in general), but under a mild technical assumption a closed form can be obtained which coincides with equation 6. This asserts that equation 6 is the natural choice for $\mathcal{F}$, modulo the choice of norm. $L_2$ norm is a natural choice due to its isotropy (direction/rotation invariance) and good empirical performance.

**Lemma 2** *The predicted* GED, *i.e.*, $\mathcal{F}_g$ *is metric. (i) Non-negativity:* $\mathcal{F}_g(\mathbf{Z}_{\mathcal{G}_\mathcal{Q}}, \mathbf{Z}_{\mathcal{G}_\mathcal{T}}) \geq 0$, *(ii) Identity:* $\mathcal{F}_g(\mathbf{Z}_{\mathcal{G}_\mathcal{Q}}, \mathbf{Z}_{\mathcal{G}_\mathcal{T}}) = 0 \iff \mathbf{Z}_{\mathcal{G}_\mathcal{Q}} = \mathbf{Z}_{\mathcal{G}_\mathcal{T}}$, *(iii) Symmetry:* $\mathcal{F}_g(\mathbf{Z}_{\mathcal{G}_\mathcal{Q}}, \mathbf{Z}_{\mathcal{G}_\mathcal{T}}) = \mathcal{F}_g(\mathbf{Z}_{\mathcal{G}_\mathcal{T}}, \mathbf{Z}_{\mathcal{G}_\mathcal{Q}})$, *(iv) Triangle Inequality:* $\mathcal{F}_g(\mathbf{Z}_{\mathcal{G}_\mathcal{Q}}, \mathbf{Z}_{\mathcal{G}_\mathcal{T}}) \leq \mathcal{F}_g(\mathbf{Z}_{\mathcal{G}_\mathcal{Q}}, \mathbf{Z}_{\mathcal{T}'}) + \mathcal{F}_g(\mathbf{Z}_{\mathcal{T}'}, \mathbf{Z}_{\mathcal{G}_\mathcal{T}})$.

PROOF. Follows trivially from the fact that $\mathcal{F}_g$ forms a *Euclidean* space. □

Similar to $\mathcal{F}$, the form for $\mathcal{F}_g$ also comes naturally from some mild technical assumptions for any embedding space function which models GED (see App. E).

**Theorem 2** *The properties of* SED *(and* GED*) in* § *2.1 are satisfied by the prediction functions* NEUROSED$(\mathcal{G}_1, \mathcal{G}_2)$ *(and the* GED *version* NEUROSED$_\mathcal{G}(\mathcal{G}_1, \mathcal{G}_2)$*).*

PROOF SKETCH. (For SED) Let $\mathcal{E}$ be the embedding function learned by NEUROSED's GNN. Then NEUROSED$(\mathcal{G}_1, \mathcal{G}_2) = \mathcal{F}(\mathcal{E}(\mathcal{G}_1), \mathcal{E}(\mathcal{G}_2))$. Now, the relevant properties follow from Lemma 1 using the substitutions $\mathbf{Z}_{\mathcal{G}_\mathcal{Q}} \mapsto \mathcal{E}(\mathcal{G}_1), \mathbf{Z}_{\mathcal{G}_\mathcal{T}} \mapsto \mathcal{E}(\mathcal{G}_2)$, and $\mathbf{Z}_{\mathcal{G}_{\mathcal{T}'}} \mapsto \mathcal{E}(\mathcal{G}_3)$. □

We note here that if different embedding models are used for $\mathcal{G}_1$ and $\mathcal{G}_2$, or if the distance computations are pair-dependent (Li et al., 2019), then Theorem 2 would not hold. It is easy to verify that the proof fails if $\mathcal{E}_\mathcal{Q}$ and $\mathcal{E}_\mathcal{T}$ are different embedding functions corresponding to the query and the target, or if $\mathcal{E}$ is a function of both $\mathcal{G}_1$ and $\mathcal{G}_2$. Hence, the siamese architecture is crucial.

**Complexity analysis:** Inference is *linear* in query and target sizes (see App. F for details).

## 3.3 Comparison with NEUROMATCH (Rex et al., 2020):

Both NEUROSED and NEUROMATCH use siamese GNNs. However, there are important differences in (1) their objectives, (2) training and (3) inference procedures and (4) theoretical characterization.

1. **Objective:** While NEUROSED is trained to predict GED and SED directly, NEUROMATCH studies the simpler task of anchored subgraph isomorphism. Thus, NEUROSED subsumes the objective of NEUROMATCH.
2. **Training:** NEUROMATCH trains the model to construct *order* embeddings wherein if $\mathcal{G}_1 \subseteq \mathcal{G}_2$, then $\forall i, \mathbf{Z}_{\mathcal{G}_1}[i] \leq \mathbf{Z}_{\mathcal{G}_2}[i]$, where $\mathbf{Z}_\mathcal{G}$ denotes the embedding of graph $\mathcal{G}$. To inject this bias, the model is trained using max-margin loss. NEUROSED does not enforce order embeddings. Rather it learns embeddings that make the proposed prediction functions (Eq. 6 and Eq. 8) effective through RMSE loss. In the context of SED, Eq. 6 is used in the loss function of both NEUROSED and NEUROMATCH. However, the loss function themselves are different, which results in the proposed embedding space being considerably richer in its ability to preserve properties from the original space. In addition, in App. E, we characterize the class of functions that may act as a substitute for Eq. 6, and thereby perform a thorough theoretical characterization of our work.

3. **Inference:** NEUROSED creates a single embedding of the query graph, based on which the SED and GED predictions are made. In contrast, NEUROMATCH generates embeddings for each node neighborhood in the query and target graph and makes $|V_Q||V_T|$ predictions corresponding to each pair of neighborhoods. The final decision of whether the query is a subgraph of the target is performed using a *voting* mechanism.

## 4 EMPIRICAL EVALUATION

In this section, we establish the following:

- **Subgraph Search:** NEUROSED outperforms the state of the art approaches for SED prediction.
- **Graph Search:** NEUROSED$_\mathcal{G}$, the variation to predict GED, outperforms the state of the art approaches for GED prediction.
- **Scalability:** NEUROSED and NEUROSED$_\mathcal{G}$ are orders of magnitude faster than existing approaches and scale well to graphs with millions of nodes.

The code base and datasets are available at https://anonymous.4open.science/r/NeuroSED/.

### 4.1 EXPERIMENTAL SETUP

Details on the hardware and software platform and parameters are provided in App. I.

**Datasets:** Table 4.1 lists the datasets used for benchmarking. Further details on the dataset semantics are provided in the App. H. We include a mixture of both *graph databases* (#graphs >1), as well as *single large* graphs (#graphs = 1). Linux and IMDB contain unlabeled graphs.

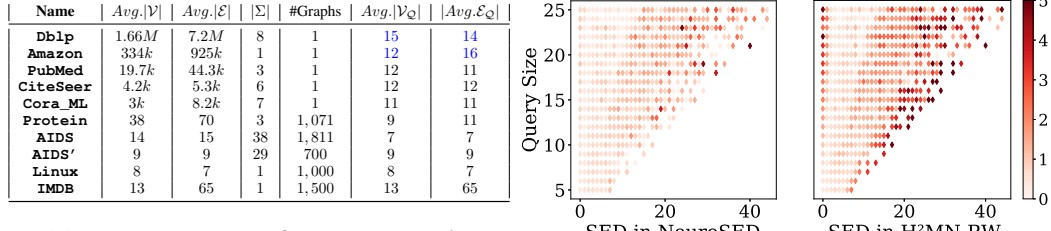

| Name | $Avg.|\mathcal{V}|$ | $Avg.|\mathcal{E}|$ | $|\Sigma|$ | #Graphs | $Avg.|\mathcal{V}_\mathcal{Q}|$ | $|Avg.\mathcal{E}_\mathcal{Q}|$ |
|---|---|---|---|---|---|---|
| Dblp | $1.66M$ | $7.2M$ | 8 | 1 | 15 | 14 |
| Amazon | $334k$ | $925k$ | 1 | 1 | 12 | 16 |
| PubMed | $19.7k$ | $44.3k$ | 3 | 1 | 12 | 11 |
| CiteSeer | $4.2k$ | $5.3k$ | 6 | 1 | 12 | 12 |
| Cora_ML | $3k$ | $8.2k$ | 7 | 1 | 11 | 11 |
| Protein | 38 | 70 | 3 | 1,071 | 9 | 11 |
| AIDS | 14 | 15 | 38 | 1,811 | 7 | 7 |
| AIDS' | 9 | 9 | 29 | 700 | 9 | 9 |
| Linux | 8 | 7 | 1 | 1,000 | 8 | 7 |
| IMDB | 13 | 65 | 1 | 1,500 | 13 | 65 |

Table 1: **Datasets used for benchmarking.**

Figure 2: **Heat Map of SED error against query size in `Dblp`. Darker means higher error.**

**Training (and Test) Data Generation:** For GED, we use $\langle query, target \rangle$ graph pairs from IMDB, AIDS', and Linux. Our setup is identical to SIMGNN (Bai et al., 2019) and H²MN-RW (Zhang et al., 2021). For SED, the procedure is more intricate. The details of datasets can be found in App. H.

- **Queries:** In single large graphs, queries are sampled by performing a random BFS traversal (depth up to 5) on a randomly chosen target graph. In addition, for AIDS, we use known *functional groups* as test queries (See App. G for details). This allows us to benchmark NEUROSED on both natural as well synthetic queries. The average query sizes ($|\mathcal{V}_\mathcal{Q}|$, $|\mathcal{E}_\mathcal{Q}|$) are listed in Table 4.1.
- **Ground-truth:** We use *mixed integer programming* method F2 (Lerouge et al., 2017) implemented in GEDLIB (Blumenthal et al., 2019) with a large time limit to generate ground-truth data.

**Train-Validation-Test Split:** We use $100K$ query-target pairs for training and $10K$ pairs each for validation and test.

**Baselines:** To evaluate performance in **GED**, we compare with SIMGNN[2] (Bai et al., 2019), GENN-A* (Wang et al., 2021) and H²MN(Zhang et al., 2021). H²MN has two versions based on random walks (H²MN-RW) and $k$-hop neighborhood (H²MN-NE). We include both.

For **SED**, no neural approaches exist. However, H²MN and SIMGNN can be trained by replacing GED with SED along with minor modifications in training. While NEUROMATCH (Rex et al., 2020) cannot predict SED, it generates a *violation score* which can be interpreted as the likelihood of the query being subgraph isomorphic to the target. The violation score can be used as a proxy for SED and used in ranking of $k$-NN ($k$-NearestNeighbour) queries. Thus, NEUROMATCH comparisons are limited to $k$-NN queries on SED. The changes required in to adapt to SED are in App. I.

---

[2]We use the PyTorch implementation of SIMGNN given on paperswithcode. The code of all other neural algorithms have been obtained from the respective authors.

| Methods | Dblp | Amazon | PubMed | CiteSeer | Cora_ML | Protein | AIDS |
|---------|------|--------|--------|----------|---------|---------|------|
| NEUROSED | **0.964** | **0.495** | **0.728** | **0.519** | **0.635** | **0.524** | **0.512** |
| H²MN-RW | 1.470 | 1.294 | 1.213 | 1.502 | 1.446 | 0.941 | 0.749 |
| H²MN-NE | 1.552 | 0.971 | 1.326 | 1.827 | 1.229 | 0.755 | 0.657 |
| SIMGNN | 1.482 | 2.810 | 1.322 | 1.781 | 1.289 | 1.223 | 0.696 |
| Branch | 2.917 | 4.513 | 2.613 | 3.161 | 3.102 | 2.391 | 1.379 |
| MIP-F2 | 3.427 | 5.595 | 3.399 | 4.474 | 3.871 | 2.249 | 1.537 |

(a) Prediction of SED.

| Methods | AIDS' | Linux | IMDB |
|---------|-------|-------|------|
| NEUROSED$_\mathcal{G}$ | **0.796** | 0.415 | **6.734** |
| H²MN-RW | 0.994 | 0.734 | 86.077 |
| H²MN-NE | 1.000 | 0.319 | 87.594 |
| GENN-A* | 0.907 | **0.267** | NA |
| SIMGNN | 1.037 | 0.666 | 66.250 |
| Branch | 3.322 | 2.474 | 6.875 |
| MIP-F2 | 2.929 | 1.245 | 82.124 |

(b) Prediction of GED

Table 2: **RMSE scores (lower is better) in (a) SED and (b) GED. GENN-A\* does not scale on graphs beyond** 10 **nodes and hence the results in `IMDB` are not reported.**

In the **non-neural** category, we use *mixed integer programming* based method MIP-F2 (Lerouge et al., 2017) with a time bound of 0.1 seconds per pair for both GED and SED. MIP-F2 provides the optimal solution given infinite time. In addition, we also compare with BRANCH (Blumenthal, 2019), which achieves an excellent trade-off between accuracy and runtime (Blumenthal et al., 2020). BRANCH uses *linear sum assignment problem with error-correction* (LSAPE) to process the search space. We use GEDLIB's (Blumenthal et al., 2019) implementation of these methods.

## 4.2 PREDICTION ACCURACY OF SED AND GED

Tables 2a and 2b present the accuracy of all techniques on SED and GED in terms of *Root Mean Square Error (RMSE)*. In App. J, we evaluate the same performance using several other metrics such as $R^2$ and Mean Absolute Error (MAE). NEUROSED outperforms all other techniques in 9 out of 10 settings. The gap in accuracy is the highest in `IMDB` for GED, where NEUROSED is more than 10 times better than the neural baselines. A deeper analysis reveals that `IMDB` graphs are significantly denser and larger than `AIDS'` or `Linux`. Thus, computing the optimal GED is harder. While all techniques have higher errors in `IMDB`, the deterioration is more severe in the baselines indicating that NEUROSED scales better with graph sizes.

To gain insight on whether the same patterns also hold in SED, in Fig. 2, we plot the *heat map* of RMSE against query graph size in `Dblp`. Specifically in this plot, each dot corresponds to a query graph $\mathcal{G}_\mathcal{Q}$. The co-ordinate of a query is $(\text{SED}(\mathcal{G}_\mathcal{Q}, \mathcal{G}_\mathcal{T}), |\mathcal{V}_\mathcal{Q}|)$. The color of a dot is the RMSE; the darker the color, the higher is the RMSE. When we compare the heat maps of NEUROSED with the most recent baseline H²MN-RW, we observe that H²MN-RW is noticeably darker. Furthermore, the dark colors are clustered on higher SED values and large query sizes (upper-right corner). This indicates that NEUROSED scales better with query size and higher SED. The heatmaps of all techniques are provided in App. L.

**Range and k-NN queries:** Range and $k$-NN queries are two of the most common database queries. We next evaluate the performance of the various algorithms on these queries. In a range query, given a distance threshold $\theta$, the goal is to retrieve all database graphs that are within $\theta$ distance from the query graph. In a $k$-NN query, given the query graph, we identify the $k$ nearest neighbors from the search space in distance-ascending order. For SED, the search space includes all target graphs of the database graphs, whereas for GED, the search space constitutes of the database graphs. The performance measures are **F1-score (Range query)** and **Kendalls's tau ($k$-NN)** (Kendall, 1938) of the predicted answer set, when compared against the ground truth.

As visible in Figs. 3a-3h, NEUROSED consistently outperforms all baselines in F1-score. Similar trend is also visible in $k$-NN queries. Specifically, in Kendall's tau (Tables 3a and 3b), barring the exception of GED in `Linux`, NEUROSED consistently scores the highest indicating best preservation of the ranking order. Overall, this shows that accurate prediction of distance also transfers to accurate querying accuracy. In App. K, we also report $k$-NN performance based on Precision@$k$. In this analysis, we do not include GENN-A\* since it is orders of magnitude slower than all neural-baselines and provides a comparable distance accuracy to H²MN. Next, we discuss efficiency in detail.

| Methods | PubMed | CiteSeer | Cora_ML | Protein | AIDS |
|---------|--------|----------|---------|---------|------|
| NEUROSED | **0.90** | **0.90** | **0.91** | **0.75** | **0.80** |
| H²MN-RW | 0.87 | 0.88 | 0.88 | 0.70 | 0.72 |
| H²MN-NE | 0.87 | 0.87 | 0.87 | 0.72 | 0.73 |
| SIMGNN | 0.85 | 0.87 | 0.86 | 0.63 | 0.73 |
| NEUROMATCH | 0.70 | 0.75 | 0.73 | 0.57 | 0.59 |

(a) Ranking in SED.

| Methods | AIDS' | Linux | IMDB |
|---------|-------|-------|------|
| NEUROSED$_\mathcal{G}$ | **0.78** | 0.90 | **0.87** |
| H²MN-RW | 0.72 | 0.89 | 0.80 |
| H²MN-NE | 0.73 | **0.93** | 0.82 |
| SIMGNN | 0.70 | 0.84 | 0.70 |

(b) Ranking in GED

Table 3: **Kendall's tau scores (higher is better).**

| Methods | Dblp | Amazon | PubMed | CiteSeer | Cora_ML | Protein | AIDS |
|---|---|---|---|---|---|---|---|
| NEUROSED | **6.84** | **1.46** | **1.30** | **1.28** | **1.25** | **0.86** | **0.84** |
| H²MN-RW | 44.68 | 23.2 | 25.79 | 27.54 | 29.04 | 19.33 | 9.63 |
| H²MN-NE | 56.82 | 40.34 | 50.64 | 54.46 | 70.59 | 28.99 | 15.76 |
| SIMGNN | 109.56 | 47.68 | 39.80 | 39.40 | 40.73 | 39.02 | 43.83 |
| BRANCH | 626.489 | 79.25 | 99.11 | 155.09 | 132.98 | 52.26 | 12.93 |
| MIP-F2 | 1979.185 | 861.95 | 606.01 | 827.65 | 790.01 | 881.77 | 360.12 |

(a) SED.

| Methods | AIDS' | Linux | IMDB |
|---|---|---|---|
| NEUROSED$_\mathcal{G}$ | **0.49** | **0.70** | **0.63** |
| H²MN-RW | 9.50 | 8.74 | 8.83 |
| H²MN-NE | 10.38 | 9.80 | 10.69 |
| GENN-A* | 12190 | 1340 | NA |
| SIMGNN | 39.21 | 38.62 | 38.98 |
| BRANCH | 10.70 | 8.24 | 127.90 |
| MIP-F2 | 593.34 | 191.88 | 1173.548 |

(b) GED

Table 4: **Running times of all methods in seconds per 10k pair (lower is better).**

| Datasets | Range ($\theta = 2$) | | | | 10-**NN** | | | |
|---|---|---|---|---|---|---|---|---|
| | CPU | | GPU | | CPU | | GPU | |
| | L-Scan | M-Tree | L-Scan | H2MN | L-Scan | M-Tree | L-Scan | H2MN |
| PubMed | 0.693 | 0.56 | 0.004 | 26.6 | 1.01 | 0.49 | 0.004 | 27.5 |
| Amazon | 9.09 | 5.07 | 0.025 | 371 | 11.3 | 4.75 | 0.027 | 372 |
| Dblp | 48 | 20.9 | 0.070 | 696 | 50.4 | 18.6 | 0.126 | 698 |

(a) Scalability

| Sampler | PubMed | CiteSeer | Amazon |
|---|---|---|---|
| BFS | 0.728 | 0.519 | 0.495 |
| RW | 0.508 | 0.770 | 0.490 |
| RWR | 0.545 | 0.754 | 0.299 |
| SHADOW | 0.966 | 0.753 | 0.830 |

(b) Query generalization

Table 5: (a) **Querying time (s) for SED in the three largest datasets. L-Scan indicates time taken by linear scan in NEUROSED (times differs based on whether executed on a CPU or in GPU). M-Tree indicates time taken by NEUROSED when indexed using an adapted Metric Tree. We only present the time of H²MN-RW since H²MN-NE is exorbitantly slow. The full table is provided in Table 9 in Appendix. (b) Generalization to unseen query distributions. BFS (seen) acts as the baseline to compare against. The numbers represent RMSE.**

### 4.3 EFFICIENCY

Table 4 presents the inference time per $10K$ graph pairs. As visible, NEUROSED is up to 1800 times faster than the non-neural baselines and up to 10 to 20 times faster than H²MN-RW, the current state of the art in GED prediction. Also note that GENN-A* is exorbitantly slow (Table 4b). GENN-A* is slower since it not only predicts the GED but also the alignment via an A* search. While the alignment information is indeed useful, computing this information across all graphs in the database may generate redundant information since an user is typically interested only on a small minority of graphs that are in the answer set. In App. K.2, we discuss this issue in detail.

**Scalability:** Here, we showcase how pair-independent embeddings, and ensuring triangle inequality leads to further boost in scalability. For this experiment, we use the three largest datasets of PubMed, Amazon and Dblp. For each dataset, we pre-compute NEUROSED embeddings of all database graphs by exploiting pair-independent embeddings. Such pre-computation is not possible in the neural or non-neural baselines. Furthermore, since the predictions of NEUROSED satisfy triangle inequality, we index the pre-computed embeddings of the database graphs using the Metric Tree (Uhlmann, 1991) index structure adapted to asymmetric distance functions. Consequently, for NEUROSED, we only need to embed the query graph and evaluate $\mathcal{F}$ to make predictions at query time. Table 5a presents the results on range and 10-NN queries. When computations are done on a GPU, NEUROSED is more than 1000 times faster than both H²MN-RW and H²MN-NE. In the absence of a GPU, H²MN is practically infeasible since expensive pair-dependent computations are done at query time. In contrast, even on a CPU, with tree indexing, NEUROSED is $\approx 50$ times faster than GPU-based H²MN-RW. Note that indexing enables up to 3-times speed-up on NEUROSED over linear scan, which demonstrates the gain from ensuring triangle inequality. These results show the benefits of using an easily computable and parallelisable prediction function $\mathcal{F}$ with theoretical guarantees.

### 4.4 ABLATION STUDY

In this study, we explore the impact of our inductive biases in learning from low-volume data. We create two variants of NEUROSED: **(1)** NEUROSED-Dual trains the two parallel GNN models separately without weight-sharing, and **(2)** NEUROSED-NN uses an MLP instead of $\mathcal{F}$. Both have strictly better representational capacity than NEUROSED, so are expected to match the performance with infinite data. Figs. 4a-4c present the results. The RMSE of NEUROSED is consistently better than NEUROSED-Dual, with the difference being more significant at low volumes. This indicates that siamese structure helps. Compared to NEUROSED, NEUROSED-NN achieves marginally better performance at larger train sizes in PubMed and CiteSeer. However, in Dblp, NEUROSED is consistently better. The number of subgraphs in a dataset grows exponentially with the node set size. Hence, an MLP needs growing training data to accurately model the intricacies of this search space. In Dblp, even 100k pairs is not enough to improve upon $\mathcal{F}$. Overall, these trends indicate that $\mathcal{F}$ enables better generalization and scalability with respect to accuracy. Furthermore, given that its

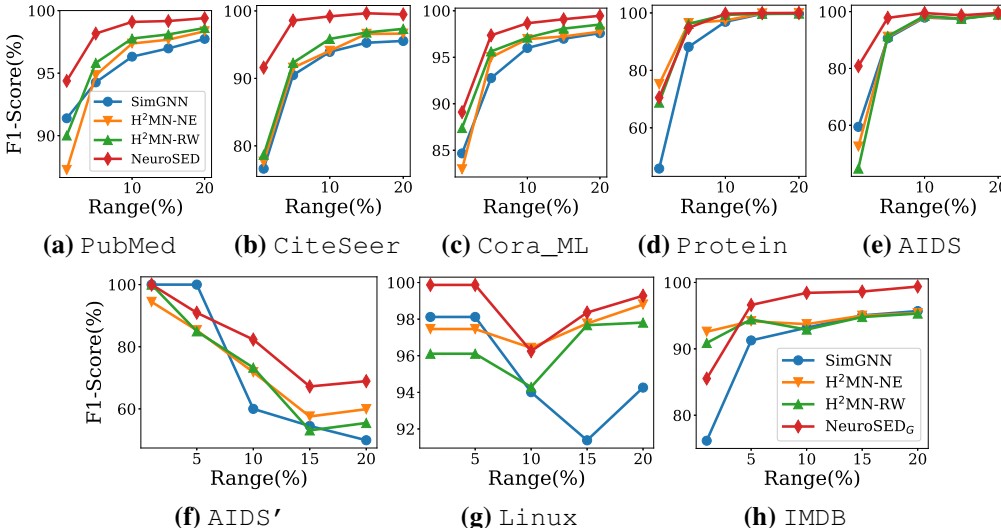

Figure 3: **F1-score in range queries on SED (a-e) and GED (f-h). The range threshold is set as a percentage of the max distance observed in the test set.**

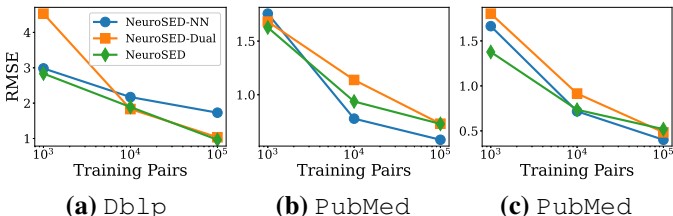

Figure 4: **Ablation study to analyze the impact of siamese architecture and function $\mathcal{F}$.**

performance is close to an MLP, and it enables indexing, the benefits outweigh the marginal reduction in accuracy. More ablations studies are provided in App. K.1.

## 4.5 GENERALIZATION TO UNSEEN QUERY DISTRIBUTIONS

We train the model by sampling queries from the graph database through BFS enumerations. *How does* NEUROSED *generalize to unseen distributions?* Towards that end, we generate queries from the three unseen distributions of **(1)** Random Walks (RW), **(2)** Random Walks with Restarts (RWR), and **(3)** SHADOW (Zeng et al., 2021). Table 5b presents the results. As visible, the errors remain low. Even more surprisingly, the errors on RW and RWR are better than the train distribution of BFS itself. This indicates good generalization to unseen distributions. See App. O for further details on the sampling strategies.

## 5 CONCLUSIONS, LIMITATIONS AND FUTURE WORK

Similarity search is a fundamental operator for graph analytics. Edit distance is one of the most popular similarity measures defined over graphs. The applicability of SED and GED, however, is constrained by its exponential computation complexity. Our work is motivated by the following question: *Is it possible to design a neural framework that generalizes to both* SED *and* GED*, preserves their key properties and is capable of learning from a low-volume training data?* We show that this is indeed possible. The efficacy and efficiency of NEUROSED stems from three algorithmic innovations: a *siamese* architecture to reduce dependence on high-volume training data, a carefully designed function to introduce an inductive bias and mimic important properties of the original space, and pair-independent embeddings to enable massive scalability.

One limitation of our work (and all neural approaches(Rex et al., 2020; Zhang et al., 2021; Wang et al., 2021; Li et al., 2019; Bai et al., 2019; 2020)) is the ability to predict on large query graph sizes. While the deterioration in prediction quality with larger query sizes is manageable if the same query sizes are seen in the train data, generalizing from small query sizes to large query sizes remain an unsolved problem (See App. M). The ability to generalize to unseen larger query sizes is important since computing GED and SED are NP-complete. Hence, generating training data on large query graphs is computationally intractable. To further advance the field of neural graph distance learning, this generalization ability .

## 6 REPRODUCIBILITY STATEMENT

Please find all code, experiments and generated data at the following anonymous link: https://anonymous.4open.science/r/NeuroSED/.

In Appendix, we have given details that are needed to make our work more reproducible. In particular, we have (1) additional proofs in App. D, (2) descriptions of datasets in App. H, (3) additional definitions in App. B, (4) analyses of computation cost for SED and GED computation in App. F, (5) additional details of the pipeline of our architecture in App. G, (6) additional details of the experimental setup in App. I, and (7) additional experimental results in App. J.

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

# APPENDIX

## A    INABILITY OF EXISTING NEURAL APPROACHES GED IN MODELING SED

A key difference between SED and GED is that while GED is symmetric, SED is not. Hence, if the architecture or training procedure relies heavily on the assumption of symmetry, then the modeling capacity is compromised.

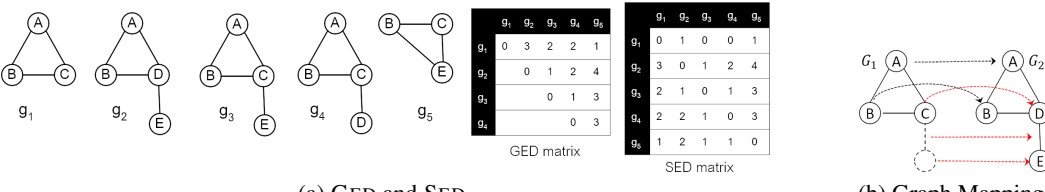

(a) GED and SED                                              (b) Graph Mapping

Figure 5: **(a) A sample set of graphs and their corresponding GED and SED matrices. (b) Example of a graph mapping. The dashed nodes and edges represent dummy nodes and edges. The red arrows denote either insertion or change of label.**

- **GMN (Li et al., 2019):** The loss function of GMN computes the euclidean distance between the embeddings and then this signal is used to learn the weight parameters (See Eq. 12 in (Li et al., 2019)). The assumption of euclidean distance prohibits the model from learning an asymmetric distance function. Certainly, the loss function can be changed to incorporate an asymmetric distance measure and the performance of the architecture following this change may be studied. However, the selection of an appropriate loss function that works well with the GMN architecture is a research problem in itself. Apart from the loss function, the proposed architecture along with the attention mechanism is also motivated by symmetry (note the quantifications for $i$ and $j$ in Eqns. (4-8) in (Li et al., 2019)).
- **GraphSim (Bai et al., 2020):** GraphSim computes a square similarity matrix across node pairs of two graphs $\mathcal{G}_1 = (\mathcal{V}_1, \mathcal{E}_1)$, $\mathcal{G}_2 = (\mathcal{V}_2, \mathcal{E}_2)$ on which CNNs are applied to predict the similarity. Since $\mathcal{G}_1$ and $\mathcal{G}_2$ may be of different sizes, the smaller graph is padded with $||\mathcal{V}_1| - |\mathcal{V}_2||$ zeros. When the similarity matrix has a large number of zeros, it confuses the model into thinking that the two graphs are dissimilar. The authors of GraphSim acknowledge this themselves while justifying a padding of $||\mathcal{V}_1| - |\mathcal{V}_2||$ zeros instead of a fixed maximum size limit. We quote an exact line from the paper:
  *"the similarity matrix between two small but isomorphic graphs may be padded with a lot of zeros, potentially misleading the CNNs to predict they are dissimilar."*
  In our problem, even if the query graph is significantly smaller than the target graph (tens of nodes vs. hundreds), SED can be 0. Thus, the above mentioned design of relying on node-to-node square similarity matrix is not well suited.
- **SimGNN (Bai et al., 2019):** SimGNN relies on a neural tensor network and a pairwise node comparison module to compute similarity between two graphs. The pairwise node comparison module assumes the comparison matrix to be symmetric. Hence, in our adaptation of SimGNN for SED, we remove this particular module and only use the neural tensor network.

## B DEFINITIONS

**Definition 5 (Graph Isomorphism)** *Graph $\mathcal{G}_1 = (\mathcal{V}_1, \mathcal{E}_1, \mathcal{L}_1)$ is isomorphic to $\mathcal{G}_2 = (\mathcal{V}_2, \mathcal{E}_2, \mathcal{L}_2)$ if there exists a bijection $\pi : \mathcal{V}_1 \to \mathcal{V}_2$ such that (i) $\forall v \in \mathcal{V}_1 : \mathcal{L}(v) = \mathcal{L}(\pi(v))$, (ii) $\forall e = (v_1, v_2) \in \mathcal{E}_1 : \pi(e) = (\pi(v_1), \pi(v_2)) \in \mathcal{E}_2$, and (iii) $\forall e \in \mathcal{E}_1 : \mathcal{L}(e) = \mathcal{L}(\pi(e))$.*

*Subgraph isomorphism* is defined analogously by using an *injection* instead of a bijection.

**Definition 6 (Nearest Subgraph)** $\mathcal{S}$ *is a nearest subgraph to* $\mathcal{G}_1$ *in* $\mathcal{G}_2$ *if* $\mathrm{SED}(\mathcal{G}_1, \mathcal{G}_2) = \mathrm{GED}(\mathcal{G}_1, \mathcal{S})$.

## C HANDLING EDGE LABELS

A straight-forward extension of NEUROSED to handle edge labels is to use GINE (Hu et al., 2019) layers instead of GIN layers. Equation 4 can be modified as follows:

$$\mathbf{h}_{v,i}^{\mathcal{G}} = \mathrm{MLP}\left( (1 + \epsilon^i) \cdot \mathbf{h}_{v,i-1}^{\mathcal{G}} + \sum_{u \in \mathcal{N}_{\mathcal{G}}(v)} \mathrm{ReLU}(\mathbf{h}_{u,i-1}^{\mathcal{G}} + \mathbf{e}_{v,u}^{\mathcal{G}}) \right) \tag{9}$$

where $\mathbf{e}_{v,u}^{\mathcal{G}}$ denotes the vector representation of the edge label for edge $(v, u)$.

It is also possible to represent edge labels in the current architecture by constructing a new graph $\mathcal{G}'$ with a vertex $w_{v,u}$ for every edge $(v, u)$ and edges from $v$ to $w_{v,u}$ and from $w_{v,u}$ to $u$. The node representation for $w_{v,u}$ is a vector representation for the label on the edge $(v, u)$ (suitably padded for uniformity). Furthermore a new dimension is introduced which separates these new nodes from the old ones and this dimension is set to 1 if the node is $w_{v,u}$ for some $(v, u) \in \mathcal{E}_{\mathcal{G}}$ and 0 otherwise. NEUROSED (and NEUROSED$_{\mathcal{G}}$) can then be trained with such modified graphs as inputs, but using the ground truth SED (respectively GED) for the original graphs in the loss function.

## D   ADDITIONAL PROOFS

### D.1   THE TRIANGLE INEQUALITY OF SED: PROOF OF THEOREM. 1

Our proof relies on two lemmas.

**Lemma 3** *Let $\widehat{d} : \Sigma \times \Sigma \to \mathbb{R}_0^+$ be a distance function over $\Sigma$, where (i) $\widehat{d}(\ell_1, \ell_2) = 0$ if $\ell_1 = \epsilon$, and (ii) $\widehat{d}(\ell_1, \ell_2) = d(\ell_1, \ell_2)$ otherwise; the following holds: $\text{SED}(\mathcal{G}_1, \mathcal{G}_2) = \widehat{\text{GED}}(\mathcal{G}_1, \mathcal{G}_2)$, where $\widehat{\text{GED}}$ denotes GED with $\widehat{d}$ as the label set distance function. In simple words, the SED between two graphs is equivalent to GED with a label set distance function where we ignore insertion costs.*

PROOF. It suffices to prove **(i)** $\text{SED}(\mathcal{G}_1, \mathcal{G}_2) \geq \widehat{\text{GED}}(\mathcal{G}_1, \mathcal{G}_2)$ and **(ii)** $\widehat{\text{GED}}(\mathcal{G}_1, \mathcal{G}_2) \geq \text{SED}(\mathcal{G}_1, \mathcal{G}_2)$.

**(i)** Let $\mathcal{S} = (\mathcal{V}_{\mathcal{S}}, \mathcal{E}_{\mathcal{S}}, \mathcal{L}_{\mathcal{S}}) \subseteq \mathcal{G}_2$ be the subgraph minimizing $\text{SED}(\mathcal{G}_1, \mathcal{S})$ (Recall Eq. 3). Consider the mapping $\pi$ from $\mathcal{G}_1$ to $\mathcal{S}$ corresponding to $\text{SED}(\mathcal{G}_1, \mathcal{S})$ (and hence $\text{GED}(\mathcal{G}_1, \mathcal{S})$ as well). We extend $\pi$ to define a mapping $\widehat{\pi}$ from $\mathcal{G}_1$ to $\mathcal{G}_2$ by mapping of all nodes in set $\mathcal{V}_2 \setminus \mathcal{V}_{\mathcal{S}}$ to dummy nodes in $\mathcal{G}_1$; the edge mappings are defined analogously.

Under this construction, $\text{SED}(\mathcal{G}_1, \mathcal{S}) = \text{GED}(\mathcal{G}_1, \mathcal{G}_2) = \widehat{\text{GED}}_{\widehat{\pi}}(\mathcal{G}_1, \mathcal{G}_2) \geq \widehat{\text{GED}}(\mathcal{G}_1, \mathcal{G}_2)$. This follows from the property that under $\widehat{d}$, insertion costs are zero, that is $\widehat{d}(\epsilon, \ell) = 0$. Thus, the additional mappings introduced in $\widehat{\pi}$ do not incur additional costs under $\widehat{d}$.

**(ii)** Consider $\mathcal{S} \subseteq \mathcal{G}_2$ and a mapping $\pi$ from $\mathcal{G}_1$ to $\mathcal{S}$ such that $\text{GED}_{\pi}(\mathcal{G}_1, \mathcal{S}) = \widehat{\text{GED}}(\mathcal{G}_1, \mathcal{G}_2)$. The existence of such a subgraph is guaranteed (See Lemma 5 in Supplementary). From the definition of GED, $\text{GED}_{\pi}(\mathcal{G}_1, \mathcal{S}) \geq \text{GED}(\mathcal{G}_1, \mathcal{S})$. Furthermore, since $\mathcal{S} \subseteq \mathcal{G}_2$, $\text{GED}(\mathcal{G}_1, S) \geq \text{SED}(\mathcal{G}_1, \mathcal{G}_2)$. Combining all these results, we have $\widehat{\text{GED}}(\mathcal{G}_1, \mathcal{G}_2) \geq \text{GED}(\mathcal{G}_1, S) \geq \text{SED}(\mathcal{G}_1, \mathcal{G}_2)$.
Hence, the claim is proved.                                                                      □

**Lemma 4** $\widehat{d}$, *as defined in Lemma 3, satisfies the triangle inequality.*

PROOF. We need to show $\widehat{d}(\ell_1, \ell_3) \leq \widehat{d}(\ell_1, \ell_2) + \widehat{d}(\ell_2, \ell_3)$. We divide the proof into four cases:
**(i)** None of $\ell_1, \ell_2, \ell_3$ is $\epsilon$. Hence, $\widehat{d}(\ell_1, \ell_3) = d(\ell_1, \ell_3)$ and the triangle inequality is satisfied.
**(ii)** $\ell_1 = \epsilon$. The LHS is 0 and hence the triangle inequality is satisfied.
**(iii)** $\ell_1 \neq \epsilon$ and $\ell_2 = \epsilon$. LHS$\leq 1$ and RHS$= 1$. Hence, satisfied.
**(iv)** Only $\ell_3 = \epsilon$. Here, LHS$= 1$ and RHS$\geq 1$.
These four cases cover all possible situations and hence, the triangle inequality is established.     □

From Lemma 3, we know $\text{SED}(\mathcal{G}_1, \mathcal{G}_2) = \widehat{\text{GED}}(\mathcal{G}_1, \mathcal{G}_2)$. Combining Obs. 3 with Lemma 4, $\widehat{\text{GED}}$ satisfies the triangle inequality. Thus, SED satisfies the triangle inequality.                                □

### D.2   PROOF OF LEMMA 5

**Lemma 5** *There exists a subgraph $\mathcal{S}$ of $\mathcal{G}_2$ and a node map $\pi$ from $\mathcal{G}_1$ to $\mathcal{S}$ such that $\text{GED}_{\pi}(\mathcal{G}_1, \mathcal{S}) = \widehat{\text{GED}}(\mathcal{G}_1, \mathcal{G}_2)$.*

PROOF. Let $\pi'$ be a node map from $\mathcal{G}_1$ to $\mathcal{G}_2$ corresponding to $\widehat{\text{GED}}(\mathcal{G}_1, \mathcal{G}_2)$. Let $h_1, \cdots, h_l$ be the nodes of $\mathcal{G}_2$ which are inserted in $\pi'$. Construct subgraph $\mathcal{S}$ of $\mathcal{G}_2$ by removing nodes $h_1, \cdots, h_l$ and their incident edges from $\mathcal{G}_2$. Let $\pi$ be the node map from $\mathcal{G}_1$ to $\mathcal{S}$ which is obtained by removing $h'_1, \cdots, h'_l$ from the domain and $h_1, \cdots, h_l$ from the co-domain of $\pi$. Since insertion costs are 0 in $\widehat{d}$ and $\pi$ contains only non-insert operations, then $\text{GED}_{\pi}(\mathcal{G}_1, \mathcal{S}) = \widehat{\text{GED}}(\mathcal{G}_1, \mathcal{G}_2)$. Hence, the claim is proved.                                                                      □

**Corollary 1** *There exists a nearest subgraph (Def. 6) $\mathcal{S}$ to $\mathcal{G}_1$ in $\mathcal{G}_2$ such that $\mathcal{S} \subseteq \mathcal{G}_1$.*

| Symbols | Descriptions |
|---|---|
| $\mathcal{V}$ | The set of nodes |
| $\mathcal{E}$ | The set of edges |
| $\mathcal{L}$ | The labeling function over nodes and edges |
| $\Sigma$ | The universe of all labels |
| $\epsilon$ | The special empty label |
| $\mathcal{L}(v)$ | The label of node $v$ |
| $\mathcal{L}(e)$ | The label of edge $e$ |
| $\pi$ | The node mapping (bijection) |
| GED | The graph edit distance |
| SED | The subgraph edit distance |
| $d$ | The distance function over the label set |
| $\mathcal{G}_{\mathcal{Q}}$ | The query graph |
| $\mathcal{G}_{\mathcal{T}}$ | The target graph |

Table 6: Frequently used symbols

PROOF. The node map $\pi$ in the above proof contains only non-insert operations. So, $\mathcal{S}$ in the above proof is a subgraph of $\mathcal{G}_1$ ignoring node and edge labels. Since equality holds in Lemma 3, this $\pi$ gives an optimal edit distance. Thus, $\mathcal{S}$ in the above proof is also a nearest subgraph to $\mathcal{G}_1$ in $\mathcal{G}_2$. $\square$

### D.3 MORE GENERAL DISTANCE FUNCTIONS

GED and SED can be viewed as error-tolerant graph and subgraph matching respectively. In practice, the edit costs are used to model the acceptable deviations from ideal graph patterns. The more tolerant we are towards a certain kind of distortion, the smaller is its corresponding edit cost. Concrete values for the edit costs are application-dependent. In this paper we have discussed binary distance functions which impose a cost of $1$ for unequal labels and $0$ for equal labels as these are the most popular distance functions used in literature. However our framework is also suitable for more general distance functions.

The results for GED directly extend to *any metric distance function* over the label set. The relevant proofs mentioned in the paper are directly applicable.

The results for SED can be extended to metric distance functions $d$ over $\Sigma$ for which replacement is no more expensive than deletion, i.e. $d(\ell_1, \ell_2) \leq d(\ell_1, \epsilon) \forall \ell_1, \ell_2 \in \Sigma$. A replacement can be seen as a compound edit operation consisting of deletion(s) followed by insertion(s). If a replacement cost is too high, replacement can be better achieved by deletion(s) followed by insertion(s), of which insertion(s) do not contribute to the cost in SED. Hence, this assumption is general enough to capture a large class of interesting distance functions which might be used for the SED prediction task. We show a generalization of Lemma 4 to such distance functions $d$. The other proofs mentioned in the paper are directly applicable.

**Lemma 6** *Let $d : \Sigma \times \Sigma \to \mathbb{R}_0^+$ be a metric and $d(\ell_1, \ell_2) \leq d(\ell_1, \epsilon) \forall \ell_1, \ell_2 \in \Sigma$. As in Lemma 3, let $\widehat{d} : \Sigma \times \Sigma \to \mathbb{R}_0^+$ be a distance function over $\Sigma$, where (i) $\widehat{d}(\ell_1, \ell_2) = 0$ if $\ell_1 = \epsilon$, and (ii) $\widehat{d}(\ell_1, \ell_2) = d(\ell_1, \ell_2)$ otherwise. Then, $\widehat{d}$ satisfies the triangle inequality.*

PROOF. We need to show $\widehat{d}(\ell_1, \ell_3) \leq \widehat{d}(\ell_1, \ell_2) + \widehat{d}(\ell_2, \ell_3)$. Consider the three cases:
**(i)** $\ell_1 = \epsilon$. LHS $= 0$, RHS $= \widehat{d}(\ell_2, \ell_3) \geq 0$ by non-negativity of $d$. Hence, LHS $\leq$ RHS.
**(ii)** $\ell_1 \neq \epsilon$ and $\ell_2 = \epsilon$. LHS $= d(\ell_1, \ell_2)$, RHS $= d(\ell_1, \epsilon)$. Hence, LHS $\leq$ RHS by assumption.
**(iii)** $\ell_1 \neq \epsilon$ and $\ell_2 \neq \epsilon$. LHS $= d(\ell_1, \ell_3)$, RHS $= d(\ell_1, \ell_2) + d(\ell_2, \ell_3)$. LHS $\leq$ RHS by triangle inequality for $d$.
These cases cover all possible situations. Hence, triangle inequality is established. $\square$

### D.4 DATA AUGMENTATION

**Observation 4** *Let $\mathcal{S}$ be a nearest subgraph (Def. 6) of $\mathcal{G}_1$ in $\mathcal{G}_2$. Let $\mathcal{S}'$ be a subgraph of $\mathcal{G}_2$ which is also a supergraph of $\mathcal{S}$. Then $\mathcal{S}$ is also a nearest subgraph of $\mathcal{G}_1$ in $\mathcal{S}'$ and $\mathrm{SED}(\mathcal{G}_1, \mathcal{S}') = \mathrm{SED}(\mathcal{G}_1, \mathcal{G}_2)$.*

PROOF. The set of subgraphs of $\mathcal{S}'$ is a subset of the set of subgraphs of $\mathcal{G}_2$. $\mathcal{S}$ achieves the minimum $\text{GED}(\mathcal{G}_1, \mathcal{S})$ among all subgraphs of $\mathcal{G}_2$ and so also the minimum $\text{GED}(\mathcal{G}_1, \mathcal{S})$ among all subgraphs of $\mathcal{S}'$. Thus, $\text{SED}(\mathcal{G}_1, \mathcal{S}') = \text{SED}(\mathcal{G}_1, \mathcal{G}_2) = \text{GED}(\mathcal{G}_1, \mathcal{S})$. □

Observation 4 gives a natural strategy for *data augmentation*. Once we have $\text{SED}(\mathcal{G}_1, \mathcal{G}_2)$ and a nearest subgraph $\mathcal{S}$ from the expensive non-neural computation of $\text{GED}(\mathcal{G}_1, \mathcal{G}_2)$, we can use $(\mathcal{G}_1, \mathcal{S}', \text{SED}(\mathcal{G}_1, \mathcal{G}_2))$ for all $\mathcal{S} \subseteq \mathcal{S}' \subseteq \mathcal{G}_2$ as training examples.

### D.5 THE TRIANGLE INEQUALITY FOR $\mathcal{F}$

This is referred to in the proof for Lemma 1.

**Lemma 7** $\mathcal{F}(\mathbf{Z}_{\mathcal{G}_{\mathcal{Q}}}, \mathbf{Z}_{\mathcal{G}_{\mathcal{T}}}) \leq \mathcal{F}(\mathbf{Z}_{\mathcal{G}_{\mathcal{Q}}}, \mathbf{Z}_{\mathcal{G}_{\mathcal{T}'}}) + \mathcal{F}(\mathbf{Z}_{\mathcal{G}_{\mathcal{T}'}}, \mathbf{Z}_{\mathcal{G}_{\mathcal{T}}})$, where $\mathcal{F}(\mathbf{x}, \mathbf{y}) = \|ReLU(\mathbf{x} - \mathbf{y})\|$ *for any monotonic norm* $\|.\|$.

PROOF. Let $\mathbf{x}, \mathbf{y} \in \mathbb{R}^n$. We observe that $(\text{ReLU}(\mathbf{x}) + \text{ReLU}(\mathbf{y}))_i = \text{ReLU}(\mathbf{x})_i + \text{ReLU}(\mathbf{y})_i = \text{ReLU}(\mathbf{x}_i) + \text{ReLU}(\mathbf{y}_i) \geq \text{ReLU}(\mathbf{x}_i + \mathbf{y}_i) = (\text{ReLU}(\mathbf{x} + \mathbf{y}))_i$. Since $\|.\|$ is monotonic, this implies $\|\text{ReLU}(\mathbf{x}) + \text{ReLU}(\mathbf{y})\| \geq \|\text{ReLU}(\mathbf{x} + \mathbf{y})\|$.

Using the triangle inequality for $\|.\|$, we get $\|\text{ReLU}(\mathbf{x})\| + \|\text{ReLU}(\mathbf{y})\| \geq \|\text{ReLU}(\mathbf{x}) + \text{ReLU}(\mathbf{y})\| \geq \|\text{ReLU}(\mathbf{x} + \mathbf{y})\|$).

Substituting $\mathbf{x} = \mathbf{Z}_{\mathcal{G}_{\mathcal{Q}}} - \mathbf{Z}_{\mathcal{G}_{\mathcal{T}'}}, \mathbf{y} = \mathbf{Z}_{\mathcal{G}_{\mathcal{T}'}} - \mathbf{Z}_{\mathcal{G}_{\mathcal{T}}}$, we get, $\|\text{ReLU}(\mathbf{Z}_{\mathcal{G}_{\mathcal{Q}}} - \mathbf{Z}_{\mathcal{G}_{\mathcal{T}'}})\| + \|\text{ReLU}(\mathbf{Z}_{\mathcal{G}_{\mathcal{T}'}} - \mathbf{Z}_{\mathcal{G}_{\mathcal{T}}})\| \geq \|\text{ReLU}(\mathbf{Z}_{\mathcal{G}_{\mathcal{Q}}} - \mathbf{Z}_{\mathcal{G}_{\mathcal{T}}})\|$

This implies that $\mathcal{F}(\mathbf{Z}_{\mathcal{G}_{\mathcal{Q}}}, \mathbf{Z}_{\mathcal{G}_{\mathcal{T}'}}) + \mathcal{F}(\mathbf{Z}_{\mathcal{G}_{\mathcal{T}'}}, \mathbf{Z}_{\mathcal{G}_{\mathcal{T}}}) \geq \mathcal{F}(\mathbf{Z}_{\mathcal{G}_{\mathcal{Q}}}, \mathbf{Z}_{\mathcal{G}_{\mathcal{T}}})$. Our claim is proved. □

## E FURTHER CHARACTERIZATION OF $\mathcal{F}$

Let $E : \mathbb{G} \to \mathbb{R}^d$, where $\mathbb{G}$ is the space of graphs, be the learned embedding function and $\mathcal{F} : \mathbb{R}^d \times \mathbb{R}^d \to \mathbb{R}$ be the embedding space prediction function. We aim to have the graph space prediction function $P : \mathbb{G} \times \mathbb{G} \to \mathbb{R}$, defined by $P(G, H) = F(E(G), E(H))$, to be able to approximate $\text{SED} : \mathbb{G} \times \mathbb{G} \to \mathbb{R}$ in the best possible way, while preserving the key theoretical properties listed below:

$$\text{SED}(G, H) \geq 0 \tag{10}$$
$$\text{SED}(G, H) = 0 \iff G \subseteq H \tag{11}$$
$$\text{SED}(G, H) \leq \text{SED}(G, T) + \text{SED}(T, H) \tag{12}$$

We allow $E$ to be arbitrary. Then, $P$ satisfies the analogous properties

$$P(G, H) \geq 0 \tag{13}$$
$$P(G, H) \leq P(G, T) + P(T, H) \tag{14}$$

if and only if $\mathcal{F}$ satisfies

$$\mathcal{F}(g, h) \geq 0 \tag{15}$$
$$\mathcal{F}(g, h) \leq \mathcal{F}(g, t) + \mathcal{F}(t, h) \tag{16}$$

For the analogue to property in Eq. 11, we first define the subgraph prediction function $P_S : \mathbb{G} \times \mathbb{G} \to \{0, 1\}$ as $P_S(G, H) = [[\forall i : E(G)[i] \leq E(H)[i]]]$. $P_S$ is motivated by NEUROMATCH (Rex et al., 2020) where a subgraph prediction function of this form satisfies the key properties of subgraph isomorphism: transitivity, anti-symmetry, intersection set and non-trivial intersection. Hence, $P$ must satisfy:

$$P(G, H) = 0 \iff P_S(G, H) = 1 \tag{17}$$

Again, $E$ being arbitrary, Eq. 17 holds if and only if, $\mathcal{F}$ satisfies:

$$\mathcal{F}(g, h) = 0 \iff \forall i : g[i] \leq h[i] \tag{18}$$

Any $\mathcal{F}$ that satisfies Eqs. 15, 16 and 18 is acceptable in our framework. However, it is not trivial to characterize such functions. To show a complete constructive characterization we add two important properties in our desiderata for $\mathcal{F}$. To motivate these properties, we first discuss about $\mathcal{F}_g$ in GED and then SED.

### E.1  GED:

For GED, we require the following three properties:

$$\mathcal{F}_g(g, h) \geq 0 \tag{19}$$

$$\mathcal{F}_g(g, h) = 0 \iff \forall i : g[i] = h[i] \tag{20}$$

$$\mathcal{F}_g(g, h) \leq \mathcal{F}_g(g, t) + \mathcal{F}_g(t, h) \tag{21}$$

With these properties in Eqs. 19, 20, and 21; $\mathcal{F}_g$ is a metric. However, giving a complete constructive characterization of metrics is not trivial. To achieve this, we establish an important connection of metrics on vector spaces to norms. Every norm $\|.\|$ gives a metric $(x, y) \mapsto \|x - y\|$. Moreover for a metric, there exists a norm $\|.\|$ such that the metric can be expressed as $(x, y) \mapsto \|x - y\|$, if and only if the metric is **translation invariant** and **homogeneous**. Thus, we add these properties to the desiderata for $\mathcal{F}_g$:

$$\mathcal{F}_g(g + k, h + k) = \mathcal{F}_g(g, h), \forall k \in \mathbb{R}^d \tag{22}$$

$$\mathcal{F}_g(rg, rh) = |r|\mathcal{F}_g(g, h), \forall r \in \mathbb{R} \tag{23}$$

**Observation 5** *The followings give a complete constructive characterization of $\mathcal{F}_g$ satisfying Eqs. 19 - 23:*

*1. $\mathcal{F}_g$ is $(x, y) \mapsto \|x - y\|$ for some norm $\|.\|$ over the vector space $\mathbb{R}^d$.*
*2. For any norm $\|.\|$ over the vector space $\mathbb{R}^d$, $(x, y) \mapsto \|x - y\|$ is a valid $\mathcal{F}_g$.*

Hence, the $L_2$ norm is a natural choice because it is **isotropic** (i.e. direction/rotation invariant).

### E.2  SED:

For SED, we add **translation invariance** and **non-negative homogeneity** to the desiderata, to make a complete constructive characterization feasible:

$$\mathcal{F}(g + k, h + k) = \mathcal{F}(g, h), \forall k \in \mathbb{R}^d \tag{24}$$

$$\mathcal{F}(rg, rh) = r\mathcal{F}(g, h), \forall r \in \mathbb{R}_0^+ \tag{25}$$

Note that non-negativity of $r$ in Eq. 25 is necessary to be consistent with Eq. 11. If $r$ is negative, then with $r = -1$, we will have $x \leq y \implies \mathcal{F}(x, y) = 0 \implies \mathcal{F}(-x, -y) = 0 \implies -x \leq -y \implies x \geq y$, which is a contradiction for $x \neq y$.

For these two properties, translation invariance and non-negative homogeneity, we provide intuitive interpretations in the context of SED. The translation invariance would imply that if graphs $G$ and $H$ are modified in the same way then the $\text{SED}(G, H)$ will be preserved. The positive homogeneity would mean that if the costs are scaled by a positive factor, then the SED is also scaled by the same positive factor.

From Eq. 24, we have $\mathcal{F}(g, h) = \mathcal{F}(g - h, 0) = p(g - h)$, where $p : \mathbb{R}^d \to \mathbb{R}$ is defined as $p(x) = \mathcal{F}(x, 0)$. Then Eqs. 16 and 25 are satisfied if and only if:

$$p(x + y) \leq p(x) + p(y) \tag{26}$$

$$p(rx) = rp(x), \forall r \in \mathbb{R}_0^+ \tag{27}$$

Here, Eq. 26 reflects subadditivity property and Eq. 27 is nonnegative homogeneity. With these properties, $p$ is an **asymmetric seminorm**. Additionally, Eq. 18 is satisfied if and only if:

$$p(x) = 0 \iff \forall i : x[i] \leq 0 \tag{28}$$

We call this Eq. 28 as **rectification** property and refer to the seminorms satisfying this property as **rectified seminorms**.

Our claim is that $\mathcal{F}$ satisfies all the desired properties presented earlier, if and only if $p$ is a rectified asymmetric seminorm. Thus, we have the following complete characterization for $\mathcal{F}$.

**Observation 6** *The followings give a complete characterization for $\mathcal{F}$:*

1. $\mathcal{F}$ is $(g, h) \mapsto p(g - h)$ for some rectified asymmetric seminorm $p$.
2. For any rectified asymmetric seminorm $p$, $(g, h) \mapsto p(g - h)$ is a valid $\mathcal{F}$.

Now, we describe the following characterization for $p$. Recall that in $d$-dimensional vector spaces, asymmetric seminorms are in one-to-one correspondence with convex sets containing the origin. Specifically, every asymmetric seminorm $p$ corresponds to a convex set $B$, containing the origin, called the **dual unit ball** of $p$. Furthermore, $p$ and $B$ are related as follows:

$$p(x) = \sup_{z \in B} z \cdot x \qquad (29)$$

Additionally, $p$ is symmetric if and only if $B$ is symmetric about the origin and $p$ is positive definite (i.e. $p(x) = 0 \implies x = 0$) if and only if $B$ contains the origin in its topological interior. Note that monotonic norms are also asymmetric seminorms.

**Lemma 8 (Rectified dual unit ball)** *Eq. 28 holds if and only if $B$ satisfies the following:*

$$\forall z \in B : \forall i : z[i] \geq 0 \qquad (30)$$

*In other words, the necessary and sufficient conditions say that $B$ lies completely in the positive orthant.*

PROOF.

1. ( $\implies$ ) If $z[i] < 0$ for some $i$ and some $z \in B$, then taking $x$ in the negative orthant, with $x[i]$ having a sufficiently large magnitude, gives positive $z \cdot x$, which leads to a positive $p(x)$, contradicting the requirement that $p(x) = 0$ when $\forall i : x[i] \leq 0$.
2. ( $\impliedby$ ) If $\forall z \in B : \forall i : z[i] \geq 0$, then for any $x$ in the negative orthant, $z \cdot x \leq 0, \forall z \in B$. $0 \in B$, so the upper bound of $0$ is achievable, hence $p(x) = 0$.

$\square$

Note that, Lemma 8 implies that the origin is at the boundary of $B$, which negates the positive definiteness of $p$. Hence, every valid $p$ can be constructed by first choosing a convex set $B$ in the positive orthant of $\mathbb{R}^d$ and then defining $p(x)$ using Eq. 29. Conversely, any such construction gives a valid $p$. However, depending on $B$, it may not be always be feasible to simplify or solve the optimization problem of Eq. 29. In the special case where $B$ is restricted to polytopes, Eq. 29 is a Linear Program. With recently proposed differentiable Linear Programming layers, $B$ could be learnt in an end-to-end fashion via backpropagation (with non-negativity constraints on the variables to enforce the positive orthant). However, this adds more parameters to learn and hyperparameters to tune, and makes training and inference significantly costlier. Instead we add a different, and arguably milder, restriction on $B$, to get a closed form for Eq. 29 in terms of norms. Specifically, we assume that $B$ is such that mirroring it on all orthants results in a convex set. Not surprisingly, this closed form coincides with the form of $\mathcal{F}$ used in the paper. This establishes that our embedding space prediction function is a natural choice for modelling SED.

First, we show a characterization for the dual unit ball of a monotonic norm. Recall that a norm $\|.\|$ on the vector space $\mathbb{R}^d$ is said to be monotonic if $(\forall i : |x'[i]| \geq |x[i]|) \implies (\|x'\| \geq \|x\|)$. This characterization allows us to construct the dual unit ball of a rectified asymmetric seminorm $p$ from the dual unit ball of a monotonic seminorm $\|.\|$, and vice versa, such that $p(x) = \|max(0, x)\|$.

**Lemma 9** *Let $\|.\|$ be a norm over $\mathbb{R}^d$. Let $B'$ be its dual unit ball. $\|.\|$ is monotonic if and only if the following holds: $\forall x : \forall i : z[i]x[i] \geq 0$, where, $z \in \arg\max_{z \in B'} z \cdot x$.*

PROOF.

1. ( $\implies$ ) Assume that $z[i]x[i] < 0$ for some $i$ and $x$. Wlog, assume $x[i] > 0$. Then, by assumption, $z[i] < 0$. Construct $x'$ by replacing $x[i]$ in $x'$ with $x[i] + \Delta$, with $\Delta > 0$. Let $z' \in \arg\max_{z' \in B'} z' \cdot x'$. By continuity, for sufficiently small $\Delta$, there is a $z'$, which is arbitrarily close to $z$. In particular, for such $\Delta$, $z'[i] < 0$. This gives $z' \cdot x' < z' \cdot x$. But monotonicity gives $z' \cdot x' \geq z \cdot x$ and maximality of $z$ for $x$ gives $z \cdot x \geq z' \cdot x$, which on chaining give $z' \cdot x' \geq z' \cdot x$. This contradicts the previous inequality.
2. ( $\impliedby$ ) Consider $x$ and $x'$ with $|x[i]| \leq |x'[i]|, \forall i$. Let $z \in \arg\max_{z \in B'} z \cdot x$, such that $z[i]x[i] \geq 0, \forall i$. Then $\|x'\| \geq z \cdot x' \geq z \cdot x = \|x\|$. Hence, $\|.\|$ is monotonic.

$\square$

**Lemma 10** *Let $\|.\|$ be a monotonic norm. Then $x \mapsto \| \max(0, x)\|$ is a rectified asymmetric seminorm.*

PROOF. Let $B'$ be the dual unit ball of a monotonic norm $\|.\|$. Let $B$ be the positive orthant of $B'$. Clearly $B$ is convex and contains the origin, so let $B$ be the dual unit ball of a rectified asymmetric seminorm $p$ (see Lemma 8). We show that $\| \max(0, x)\| = p(x)$, which completes the proof.

$$\| \max(0, x)\| = \sup_{z \in B'} z \cdot \max(0, x)$$
$$= \sup_{z \in B} z \cdot max(0, x)[\max(0, x) \geq 0]$$
$$= \sup_{z \in B} z \cdot x$$
$$= p(x)$$

Here, $\max(0, x) \geq 0$, as by Lemma 9 it suffices to consider $z$'s in the positive orthant, where $B$ coincides with $B'$. Additionally, for $x[i] < 0$ as the supremum is at $z[i] = 0$; $\max(0, x)$ coincides with $x$ at $i : x[i] \geq 0$. $\square$

**Lemma 11** *Let $p$ be a rectified asymmetric seminorm with a dual unit ball $B$ such that mirroring $B$ on all orthants results in a convex set. Then $p$ is $x \mapsto \| \max(0, x)\|$ for some monotonic norm $\|.\|$.*

PROOF. Let $B$ be the dual unit ball of $p$. Let $B' = \{z : |z| \in B\}$. $B'$ is convex (by mirroring assumption), symmetric about the origin, and contains the origin in its topological interior (except in degenerate cases which we ignore), so $B'$ is the dual unit ball for a norm $\|.\|$. For any $x$, let $z \in \arg\max_{z \in B'} z \cdot x$, and consider $z'$ obtained from $z$ by keeping the magnitudes of $z[i]$'s but taking the sign of $x[i]$'s. By construction $z' \in B'$, and $z' \cdot x \geq z \cdot x$, which gives $z' \in \arg\max_{z \in B'} z \cdot x$. So by Lemma 9, $\|.\|$ is a monotonic norm. Now, $B$ is the positive orthant of $B'$, so by the reasoning of Lemma 10, $p(x) = \| \max(0, x)\|$, and it completes the proof. $\square$

**Discussion:** $L_p$ norms are natural choices for the monotonic norms $\|.\|$. For $L_1$ norm, the dual unit ball is a hypercube with side length 2 units and centered at the origin. For $L_2$ norm, the dual unit ball is a hypersphere of unit radius centered at the origin. The dual unit balls of the corresponding rectified asymmetric seminorms are simply the positive orthants of the hypercube and the hypersphere respectively, which easily satisfy the mirroring assumption. We choose $L_2$ norm because of it's isotropy, although other $L_p$ norms are also likely to work as well as the $L_2$ norm. In fact, any rectified asymmetric seminorm whose dual unit ball is obtained by taking the positive orthant of a monotonic norm's dual unit ball will satisfy the mirroring assumption, and will therefore be expressible in the form $\| \max(0, x)\|$.

## F   COMPUTATION COST OF SED AND GED INFERENCE

For this analysis, we make the simplifying assumption that the hidden dimension in the Pre-MLP, GIN and Post-MLP are all $d$. The average density of the graph is $g$. The number of hidden layers in Pre-MLP, and Post-MLP are $L$, and $k$ in GIN.

The computation cost per node for each of these components are as follows.

- **Pre-MLP:** The operations in the MLP involve linear transformation over the input vector $\mathbf{x}_v$ of dimension $|\Sigma|$, followed by non-linearity. This results in $O(|\mathcal{V}|(|\Sigma| \cdot d + d^2 L))$ cost.
- **GIN:** GIN aggregates information from each of the neighbors, which consumes $O(d \cdot g)$ time. The linear transformation consumes an additional $O(d^2)$ time. Applying non-linearity takes $O(d)$ time since it is a linear pass over the hidden dimensions. Finally these operations are repeated over each of the $k$ hidden layers, results in a total $O(k(d^2 + dg))$ computation time per node. Across, all nodes, the total cost is $O(|\mathcal{V}|kd^2 + |\mathcal{E}|kd)$ time. The degree $g$ terms gets absorbed since each edge passes message twice across all nodes.
- **Concatenation:** This step consumes $O(kd)$ time per node.
- **Pool:** Pool iterates over the GIN representation of each node requiring $O(|\mathcal{V}|dk)$ time.
- **Post-MLP:** The final MLP takes $dk$ dimensional vector as input and maps it to a $d$ dimensional vector over $L$ layers. This consumes $O(kd^2 + d^2 L)$ time.

Combining all these factors, the total inference complexity for a graph is $O(|\mathcal{V}|(|\Sigma| \cdot d + d^2L + kd^2) + |\mathcal{E}|kd)$. This operation is repeated on both the query and target graphs to compute their embeddings, on which distance function $\mathcal{F}$ is operated. Thus, the final cost is $O(n(|\Sigma| \cdot d + d^2L + kd^2) + mkd)$, where $n = |\mathcal{V}_\mathcal{Q}| + |\mathcal{V}_\mathcal{T}|$ and $m = |\mathcal{E}_\mathcal{Q}| + |\mathcal{E}_\mathcal{T}|$.

## G   PIPELINE DETAILS

**Additional Model Details:** We use residual connections across blocks of two GIN convolution layers. This eases the flow of gradients to the earlier layers by skipping the intermediate non-linearities. We observed about 5x speedup in convergence and a marginal improvement in the generalization error with residual connections. We use $8$ layers, $64$ hidden dimensions and $64$ embedding dimensions. We use a single layer for pre-MLP and $2$ layers with ReLU non-linearity for the other MLP's.

**Data generation:** We generate training data under two different settings (described in § 4). SED computation is NP-hard, and to the best of our knowledge, there is no existing method which computes SED directly. Therefore, we generate the ground truth SED using existing methods on GED by leveraging Lemma 3 which establishes the relationship between SED and GED. More specifically, we use time limited *mixed integer programming* method MIP-F2 (Lerouge et al., 2017), which provides us with the best so far lower bound and upper bound for SED. The time limit is kept at $60$ seconds per pair. Each pair is run with $64$ threads on a $64$ core machine. This gives sufficiently tight bounds (with exact values for a large majority of pairs). For evaluation, we use $\text{SED} = (\text{LB} + \text{UB})/2$ as the ground truth, where LB and UB are the lower and upper bounds respectively. Fig. 6 presents a visualization of tightness of SED bounds.

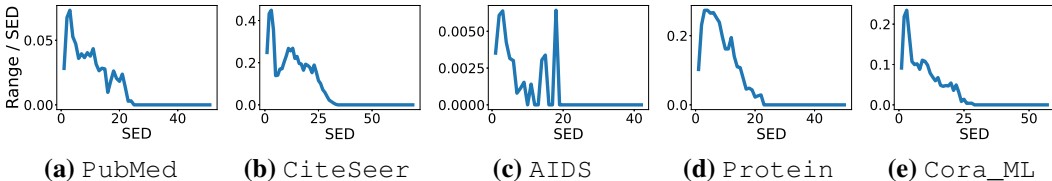

| **(a)** PubMed | **(b)** CiteSeer | **(c)** AIDS | **(d)** Protein | **(e)** Cora_ML |

Figure 6: Mean Relative Error in Generated Data. Here Range $= \text{UB} - \text{LB}$, and $\text{SED} = (\text{LB} + \text{UB})/2$.

**Real queries for AIDS:** The queries for AIDS constitute of known functional groups. They have been compiled from Table 1 in ''*Mining statistically significant molecular substructures for efficient molecular classification S Ranu, AK Singh - Journal of chemical information and modeling, 2009.*''

**Loss Function:** While $\mathbf{Z}_{\mathcal{G}_\mathcal{Q}}$ and $\mathbf{Z}_{\mathcal{G}_\mathcal{T}}$ be the embeddings for the query graph $\mathcal{G}_\mathcal{Q}$ and target graph $\mathcal{G}_\mathcal{T}$ respectively, we use the following function (for training):

$$H(\mathbf{Z}_{\mathcal{G}_\mathcal{Q}}, \mathbf{Z}_{\mathcal{G}_\mathcal{T}}, l, u) = \begin{cases} (\mathcal{F}(\mathbf{Z}_{\mathcal{G}_\mathcal{Q}}, \mathbf{Z}_{\mathcal{G}_\mathcal{T}}) - u)^2, & \mathcal{F}(\mathbf{Z}_{\mathcal{G}_\mathcal{Q}}, \mathbf{Z}_{\mathcal{G}_\mathcal{T}}) > u \\ 0, & l \leq \mathcal{F}(\mathbf{Z}_{\mathcal{G}_\mathcal{Q}}, \mathbf{Z}_{\mathcal{G}_\mathcal{T}}) \leq u \\ (l - \mathcal{F}(\mathbf{Z}_{\mathcal{G}_\mathcal{Q}}, \mathbf{Z}_{\mathcal{G}_\mathcal{T}}))^2, & \mathcal{F}(\mathbf{Z}_{\mathcal{G}_\mathcal{Q}}, \mathbf{Z}_{\mathcal{G}_\mathcal{T}}) < l \end{cases}$$

where $l$ and $u$ are the lower and upper bounds to SED in our generated data. With $H$, the parameters of the entire model are learned by minimizing the mean squared error.

$$\mathscr{L} = \frac{1}{|\mathbb{T}|} \sum_{\forall \langle \mathcal{G}_\mathcal{Q}, \mathcal{G}_\mathcal{T}, l, u \rangle \in \mathbb{T}} H(\mathbf{Z}_{\mathcal{G}_\mathcal{Q}}, \mathbf{Z}_{\mathcal{G}_\mathcal{T}}, l, u) \tag{31}$$

Here, $\mathbb{T}$ is the set of all training samples.

**Training details:** NEUROSED, NEUROSED-NN and NEUROSED-Dual were trained using the *AdamW* optimizer with a weight decay of $10^{-3}$. A batch-size of $200$ graph pairs per mini-batch was used. The learning rate schedule was *cyclic*. In each cycle, learning rate was increased from $0$ to $10^{-3}$ for 2000 iterations and then decreased back to $0$ for 2000 iterations. Training was stopped when the best validation loss did not improve for $5$ cycles. We observe that a cyclic learning rate schedule usually gives faster and better convergence than the other attempted training schemes. A cyclic learning rate schedule allows using higher learning rates and thus speeds up training. It offers some regularization due to the high learning rates seen in each cycle encouraging the optimizer to find flatter regions of minima, giving better generalization. It is also robust to hyper-parameter settings.

**Subgraph search via neighborhood decomposition:**

For large target graphs, it is infeasible to compute the SED. In this paper, we proposed a *neighborhood decomposition* scheme to handle subgraph similarity queries in the following setting: compute the SED between the query and the $k$-hop neighborhood centered at each node in the target graph for some suitable $k$. The minimum SED thus computed can be considered as an approximation for the true SED. Ideally, such an SED would be an upper bound to the true SED. In our experiments the minimum SED is usually small. This is expected in general as the large targets graphs are likely to contain small query graphs (or their small variations). Thus the neighborhood decomposition is satisfactory for SED computation between small queries and large targets in practice. One practical advantage of using neighborhood decomposition is that it facilitates locating the region of the target graph where the nearest subgraph is found. We rank these neighbourhoods and that allows us to consider regions of the target graph in order of relevance. This generates much smaller candidate sets and traditional expensive methods can be used to refine these candidate sets further.

In this section, we seek to give further insight into the efficacy of neighborhood decomposition and design guidelines for choosing the right value of $k$ depending on the expected query distribution. In practice, we are concerned only with connected subgraphs. Assuming that all the nearest subgraphs (Def. 6) are connected leads to the following result:

**Theorem 3** *Assuming that the nearest subgraph to the query is a connected graph, a nearest subgraph $\mathcal{S}$ to $\mathcal{G}_1$ in $\mathcal{G}_2$ can be found in an $l/2$-hop neighborhood $\mathcal{N}_v$, centered at some $v \in \mathcal{V}_{\mathcal{G}_2}$, where $l$ is the length of the longest path in $\mathcal{G}_1$.*

PROOF. By Corollary 1, there exists a nearest subgraph $\mathcal{S}$ of $\mathcal{G}_1$ in $\mathcal{G}_2$ which is also a subgraph of $\mathcal{G}_1$ ignoring node and edge labels. So, the diameter of $\mathcal{S}$ is $\leq$ the length $l$ of the longest path in $\mathcal{G}_1$. Since $\mathcal{S} \subseteq \mathcal{G}_2$, $\mathcal{S}$ is contained in some $l/2$ neighborhood of $\mathcal{G}_2$. $\square$

Finding the length of the longest path is NP-hard. However, we do not need to find the exact length of the longest path: any upper bound suffices. Moreover we do not even need the length of the longest path for $l$. The nearest subgraph having a diameter equal to $l$ is rare. In practice, the diameter of the nearest subgraph is unlikely to be much higher than the diameter of the query itself.

In conclusion, subgraph similarity search suffers from the problem of an exponential search space of subgraphs. We resolve this issue in two ways: the neighborhood decomposition allows us to consider a number of subgraphs linear in the number of nodes in the target, all of which are much smaller than the target graph itself. Second, our neural model learns to predict the SED directly with the small neighborhood sizes which make training data generation as well as neural processing using GNNs computationally feasible.

## H  DATASETS

**Dblp:** `Dblp` is a co-authorship network where each node is an author and two authors are connected by an edge if they have co-authored a paper. The label of a node is the venue where the authors has published most frequently. The dataset has been obtained from https://www.aminer.org/citation.

**Amazon:** Each node in `Amazon` represents a product and two nodes are connected by an edge if they are frequently co-purchased. The graph is unlabeled and hence equivalent to a graph containing a single label on all nodes. The dataset has been downloaded from (Leskovec & Sosič, 2016).

**PubMed:** `PubMed` dataset is a citation network which consists of scientific publications from `PubMed` database pertaining to diabetes classified into one of three classes.

**Protein:** `Protein` dataset consists of protein graphs. Each node is labeled with a one of three functional roles of the protein.

**AIDS:** `AIDS` dataset consists of graphs constructed from the AIDS antiviral screen database. These graphs representing molecular compounds with Hydrogen atoms omitted. Atoms are represented as nodes and chemical bonds as edges.

**CiteSeer:** `CiteSeer` is a citation network which consists of scientific publications classified into one of six classes. Generally a smaller version is used for this dataset, but we use the larger version from (Bojchevski & Günnemann, 2017).

**Cora_ML:** Cora dataset is a citation dataset consisting of many scientific publications classified into one of seven classes based on paper topic. `Cora_ML` is a smaller datset extracted from Cora (Bojchevski & Günnemann, 2017).

**AIDS':** `AIDS'` dataset is another collection of graphs constructed from the AIDS antiviral screen database. The graphs and their properties differ from those in `AIDS`. These graphs also represent chemical compound structures.

**Linux:** `Linux` dataset is a collection of program dependence graphs, where each graph is a function and nodes represent statements while edges represent dependency between statements.

**IMDB:** `IMDB` dataset is a collection of ego-networks of actors/actresses that have appeared together in any movie.

We use the same versions of `AIDS'`, `Linux`, and `IMDB` as used in (Bai et al., 2019).

## I    EXPERIMENTAL SETUP

**Hardware Details:** We use a machine with an Intel Xeon Gold 6142 processor and GeForce GTX 1080 Ti GPU for all our experiments.

**Baselines:** Here we describe the changes made to neural baselines to adapt to SED problem.

- SIMGNN (Bai et al., 2019): We train the model with $\frac{\text{LB}+\text{UB}}{2}$ as the ground truth value of SED (LB and UB are lower and upper bounds for SED respectively). Now let $\mathcal{G}_1$ be the query graph, and $\mathcal{G}_2$ be the target graph, we use $n\text{SED}(\mathcal{G}_1, \mathcal{G}_2) = \frac{\text{SED}(\mathcal{G}_1, \mathcal{G}_2)}{|\mathcal{G}_1|}$ instead of $n\text{GED}(\mathcal{G}_1, \mathcal{G}_2) = \frac{\text{GED}(\mathcal{G}_1, \mathcal{G}_2)}{(|\mathcal{G}_1| + |\mathcal{G}_2|)/2}$ as the normalized value of similarity in SIMGNN. This is because the absolute value of SED scales with the size of the query, as opposed to GED, which scales with size of the query as well as the target.
- NEUROMATCH (Rex et al., 2020): We use the *un-anchored* version of NEUROMATCH trained on its original task (subgraph isomorphism decision problem) using its own training framework. For ranking, we use the violation scores it produces directly rather than thresholding the violation and predicting subgraph isomorphism as intended. We change the size limits for sampling query/target graphs in training to better fit our data distribution.

**Querying for SED:** We embed the neighbourhoods of the nodes in the given graph (or set of graphs) and store the embeddings (denoted by set $\mathbf{Z}_{\mathcal{G}_{\mathcal{T}}}$) accordingly. For a new query $G_Q$, we first obtain its embedding $\mathbf{Z}_{\mathcal{G}_{\mathcal{Q}}}$. Based on the stored embeddings $\mathbf{Z}_{\mathcal{G}_{\mathcal{T}}}$, we compute $\text{SED}(\mathbf{Z}_{\mathcal{G}_{\mathcal{Q}}}, \mathbf{Z}_{\mathcal{G}_{\mathcal{T}}})$ for all $\mathbf{Z}_{\mathcal{G}_{\mathcal{T}}} \in \mathbf{Z}_{\mathcal{G}_{\mathcal{T}}}$ using the function $\mathcal{F}(\mathbf{Z}_{\mathcal{G}_{\mathcal{Q}}}, \mathbf{Z}_{\mathcal{G}_{\mathcal{T}}})$. We sort the targets based on the obtained SED values and it can trivially answer top-k or range queries. From the selected targets one can extract the nearest subgraph by running a traditional algorithm to compute SED and use the returned node mapping to interpret it. An alternative is to choose a larger ($k'$, where $k' > k$) top set and use traditional methods to further filter these. Note that usually the graphs required in the top-k or range queries are much smaller than the total number of targets, so using our model to filter out the unimportant targets from the target candidate set can save significant amount of resources.

**Software details:** We used Pytorch and Pytorch-Geometric for implementation. All experiments were done using Jupyter Notebooks. The full code, notebooks for all experiments and the generated data are available on an anonymous github repository [3].

**License:** The provided code and data are licensed under **CC-BY-SA**.

## J    ADDITIONAL RESULTS

### J.1    OTHER MEASURES FOR PREDICTING SED AND GED

In Section 4.2, we present the accuracy of all techniques on SED and GED in terms of *Root Mean Square Error (RMSE)*. Here we evaluate the same performance using several other metrics such as Mean Absolute Error (MAE) and $R^2$. Tables 7a and 7b show the MAE scores for SED and GED respectively. Tables 8a and 8b present the results based on $R^2$ results. Consistent with the previous results, NEUROSED outperforms all techniques both in SED and GED in these two measures except

---
[3] `https://anonymous.4open.science/r/NeuroSED/`

| Methods | PubMed | Protein | AIDS | CiteSeer | Cora_ML | Amazon | Dblp ‖ |
|---|---|---|---|---|---|---|---|
| NEUROSED | **0.474** | **0.377** | **0.401** | **0.332** | **0.432** | **0.207** | **0.626** ‖ |
| H²MN-RW | 0.764 | 0.575 | 0.566 | 0.882 | 0.824 | 0.480 | 0.926 ‖ |
| H²MN-NE | 0.831 | 0.518 | 0.510 | 1.094 | 0.797 | 0.465 | 0.966 ‖ |
| SIMGNN | 0.853 | 0.835 | 0.545 | 1.068 | 0.841 | 1.257 | 0.911 ‖ |
| Branch | 1.692 | 1.986 | 1.092 | 2.275 | 2.277 | 3.697 | 2.156 ‖ |
| MIP-F2 | 1.528 | 1.264 | 0.684 | 1.940 | 1.834 | 3.303 | 2.165 ‖ |

(a) Prediction of SED.

| Methods | AIDS' | Linux | IMDB ‖ |
|---|---|---|---|
| NEUROSED$_\mathcal{G}$ | **0.629** | 0.318 | 3.612 ‖ |
| H²MN-RW | 0.777 | 0.534 | 28.486 ‖ |
| H²MN-NE | 0.779 | **0.176** | 28.182 ‖ |
| SIMGNN | 0.816 | 0.489 | 28.082 ‖ |
| Branch | 2.895 | 2.017 | **3.303** ‖ |
| MIP-F2 | 2.184 | 0.480 | 36.460 ‖ |

(b) Prediction of GED

Table 7: **MAE scores (lower is better) in (a) SED and (b) GED.**

| Methods | PubMed | Protein | AIDS | CiteSeer | Cora_ML | Amazon | Dblp ‖ |
|---|---|---|---|---|---|---|---|
| NEUROSED | **.99** | **.98** | **.98** | **.99** | **.99** | **.99** | **.97** ‖ |
| H²MN-RW | .97 | .94 | .95 | .98 | .97 | .97 | .94 ‖ |
| H²MN-NE | .97 | .96 | .96 | .97 | .98 | .98 | .93 ‖ |
| SIMGNN | .97 | .9 | .95 | .97 | .97 | .85 | .94 ‖ |
| Branch | .88 | .64 | .82 | .91 | .86 | .63 | .75 ‖ |
| MIP-F2 | .80 | .68 | .78 | .83 | .79 | .44 | .65 ‖ |

(a) Prediction of SED.

| Methods | AIDS' | Linux | IMDB ‖ |
|---|---|---|---|
| NEUROSED$_\mathcal{G}$ | **.91** | .97 | **.99** ‖ |
| H²MN-RW | .86 | .92 | .66 ‖ |
| H²MN-NE | .86 | **.98** | .65 ‖ |
| SIMGNN | .84 | .93 | .80 ‖ |
| Branch | -.55 | .14 | .98 ‖ |
| MIP-F2 | -.20 | .78 | .69 ‖ |

(b) Prediction of GED

Table 8: $R^2$ **scores (higher is better, $1$ is maximum) in (a) SED and (b) GED.**

BRANCH is competitive only on IMDB for GED prediction and H²MN-NE is better in Linux only for GED prediction.

## K  PRECISION@$k$ ON $k$-NN QUERIES

Fig. 7 shows Precision@$k$ on $k$-NN queries. As visible, either NEUROSED comfortably performs better than all baselines or is among the best performing algorithms.

### K.1  ADDITIONAL RESULTS ON ABLATION STUDY

**Impact of GIN:** To highlight the importance of GIN, we conduct ablation studies by replacing the GIN convolution layers in the model with several other convolution layers. As visible in the Table 10, GIN consistently achieves the best accuracy. This is not surprising since GIN is provably the most expressive among GNNs in distinguishing graph structures (essential to SED or GED computation) and is as powerful as the Weisfeiler-Lehman Graph Isomorphism test (Xu et al., 2018b).

| Methods | CiteSeer (SED) | PubMed (SED) | Amazon (SED) | IMDB (GED) |
|---|---|---|---|---|
| **NEUROSED (GIN)** | **0.519** | **0.728** | **0.495** | **6.734** |
| **NEUROSED-GCN** | 0.556 | 0.756 | 0.532 | 12.151 |
| **NEUROSED-GRAPHSAGE** | 1.364 | 1.156 | 1.841 | 91.312 |
| **NEUROSED-GAT** | 1.294 | 1.259 | 1.843 | 89.034 |

Table 10: **Ablation studies: GIN vs others. RMSE produced by different methods are shown and NEUROSED with GIN produces the best results.**

**Impact of sum-pool:** To substantiate our choice of the pooling layer, we have performed ablation studies with various pooling functions as replacements for sum-pool. It is clear from Table 11 that sum-pool is the best choice among the considered alternatives.

| Datasets | Range ($\theta = 2$) | | | | | 10-NN | | | | |
| --- | --- | --- | --- | --- | --- | --- | --- | --- | --- | --- |
| | CPU | | GPU | | | CPU | | GPU | | |
| | L-Scan | M-Tree | L-Scan | H²MN-RW | H²MN-NE | L-Scan | M-Tree | L-Scan | H²MN-RW | H²MN-NE |
| PubMed | 0.693 | 0.56 | 0.004 | 26.6 | 35.2 | 1.01 | 0.49 | 0.004 | 27.5 | 35.5 |
| Amazon | 9.09 | 5.07 | 0.025 | 371 | 8550 | 11.3 | 4.75 | 0.027 | 372 | 8760 |
| Dblp | 48 | 20.9 | 0.070 | 696 | 8910 | 50.4 | 18.6 | 0.126 | 698 | 9790 |

Table 9: **Querying time (s) for SED in the three largest datasets. L-Scan indicates time taken by linear scan in NEUROSED (times differs based on whether executed on a CPU or in GPU). M-Tree indicates time taken by NEUROSED when indexed using an adapted Metric Tree.**

| Pool functions | CiteSeer (SED) | PubMed (SED) | Amazon (SED) | IMDB (GED) |
| --- | --- | --- | --- | --- |
| **NEUROSED (Sum)** | **0.519** | 0.728 | **0.495** | **6.734** |
| **NEUROSED-Max** | 0.795 | **0.709** | 0.603 | 52.519 |
| **NEUROSED-Mean** | 0.922 | 0.732 | 0.846 | 52.483 |
| **NEUROSED-Attention** | 0.914 | 0.797 | 0.868 | 130.47 |

Table 11: **Ablation studies: sum-pool vs others. The sum-pool is the best choice among the considered alternatives.**

## K.2 ALIGNMENT

In real-world applications of subgraph similarity search, alignments are of interest only for a small number of similar subgraphs. Our framework is intended to serve as a filter to retrieve this small set of similar subgraphs from a large number of candidates. To elaborate, a graph database may contain thousands or millions of graphs (or alternatively, thousands or millions of neighborhoods of a large graph) which need to be inspected for similar subgraphs. A user is typically interested in only a handful of these subgraphs that are highly similar to the query. Since the filtered set is significantly smaller, a non-neural exact algorithm suffices to construct the alignments (Lemma 1 allows us to adapt general cost GED alignment techniques for SED alignment). Computing alignments across the entire database is unnecessary and slows down the query response time.

To substantiate our claim, we show the average running time for answering 10-NN queries. We break up the running time into two components: (i) $k$-NN retrieval time by NEUROSED, (ii) exact alignment time using MIP-F2 for the 10-NN neighborhoods retrieved by NEUROSED. We observe that exact alignment by existing methods on the 10-NN neighborhoods completes in reasonable time. In contrast, GENN-A$^*$ does not scale on either PubMed or Amazon since it computes alignments across *all* (sub)graphs.

| Methods | PubMed | Amazon |
| --- | --- | --- |
| **NEUROSED Retrieval** | 0.373 | 6.471 |
| **MIP-F2 Alignment** | 52.8 | 68.4 |

Table 12: **The average running times in seconds per top-10 query. Our technique is much faster than MIP-F2 alignment.**

## L  HEAT MAPS FOR PREDICTION ERROR

In Figures 8 to 17, we show the variation of the errors on SED and GED prediction with query sizes and ground truth values for NEUROSED (NEUROSED$_\mathcal{G}$ for GED) and the baselines on all the corresponding datasets. This experiment is an extension of the heat-map results in Fig. 2 in the main paper. These datasets show variations in the distributions of SED and GED values. It is interesting to observe that among the baselines, different methods perform well on different regions (e.g., combinations of high/low SED and high/low query sizes). In the figures, darker means higher error. The baselines do not show good performance on all regions. However, for NEUROSED and NEUROSED$_\mathcal{G}$, we see a much better coverage for all types of regions in the domain. Furthermore, for every region, our models outperform (or are at least competitive with) the best performing baseline for that region.

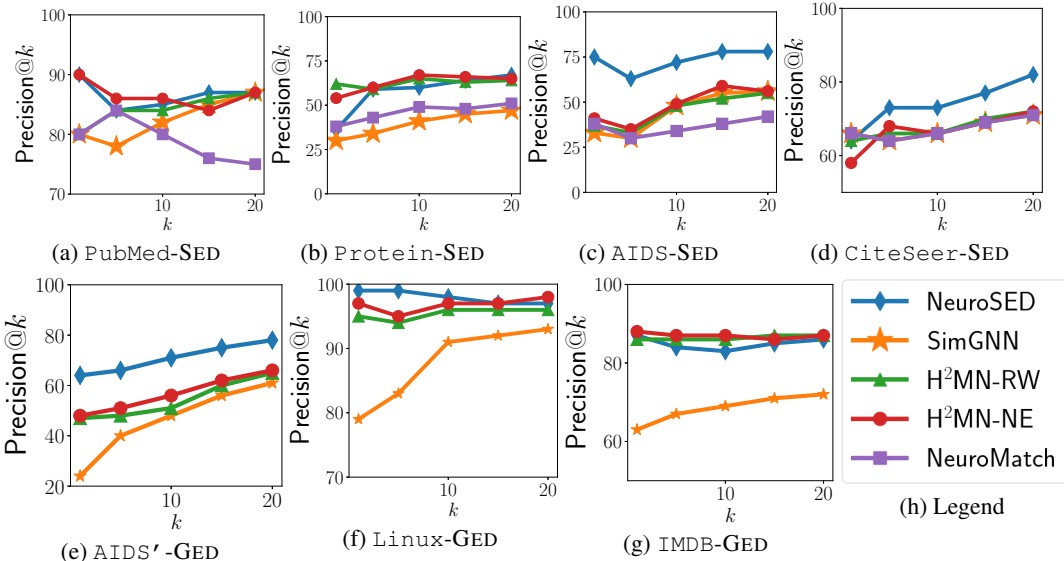

Figure 7: Precision@$k$ on $k$-NN queries.

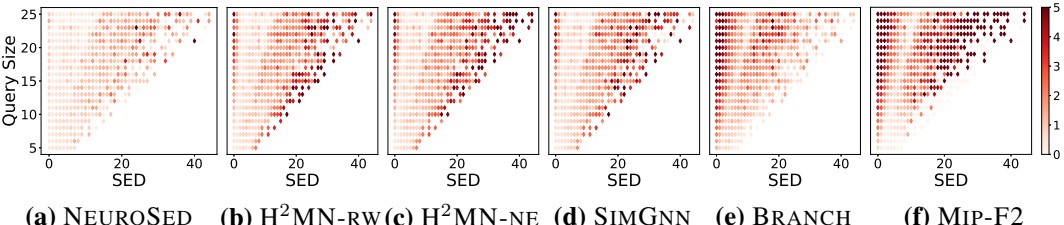

Figure 8: **Heat Maps of SED error against query size and SED values for `Dblp`. Darker means higher error.**

## M SCALABILITY WITH QUERY SIZE AND GENERALIZABILITY TO UNSEEN LARGER QUERY SIZES

In this analysis, we aim to analyze the following aspects:

1. Performance on larger query sizes.
2. Generalizability to unseen larger query sizes

In Table 13 sheds light on the above objectives. We notice that although there is some deterioration in the quality for query sizes in the range $[25, 50]$ when compared to the entire set, it is not severe (NEUROSED-50 in Table 13). However, if we the train set only contains queries till size 25 and we deploy the learned model to infer on queries of larger unseen sizes, the drop in quality is significant (NEUROSED-25 in Table 13). Nonetheless, NEUROSED remains superior to the optimal non-neural approach (MIP-F2) when run with a generous time limit of 60 seconds per query. Overall, this experiment highlights one direction that needs further study and improvement.

| Method | PubMed $\mathcal{V}_Q \in [0, 50]$ | PubMed $\mathcal{V}_Q \in [25, 50]$ | CiteSeer $\mathcal{V}_Q \in [0, 50]$ | CiteSeer $\mathcal{V}_Q \in [25, 50]$ | Amazon $\mathcal{V}_Q \in [0, 50]$ | Amazon $\mathcal{V}_Q \in [25, 50]$ |
|---|---|---|---|---|---|---|
| NEUROSED-50 | 1.294 | 1.917 | 0.728 | 0.948 | 0.638 | 0.782 |
| NEUROSED-25 | 2.824 | 4.999 | 4.740 | 9.052 | 1.152 | 1.724 |
| MIP-F2 | 3.507 | 6.278 | 4.831 | 8.505 | 6.454 | 10.293 |

Table 13: RMSE of prediction quality. NEUROSED-50 indicates NEUROSED trained on a dataset containing queries of size up to $50$. NEUROSED-25 is defined analogously.

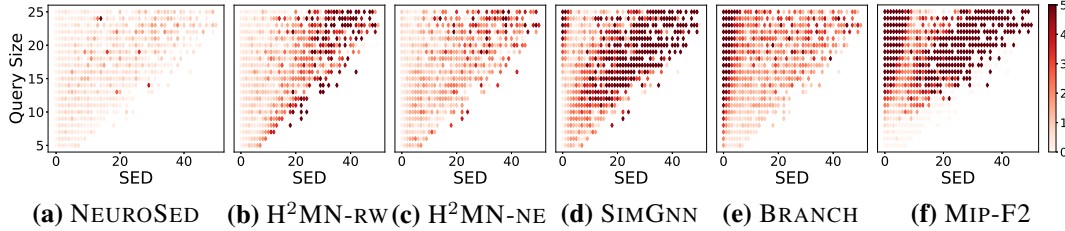

**(a)** NEUROSED **(b)** H$^2$MN-RW **(c)** H$^2$MN-NE **(d)** SIMGNN **(e)** BRANCH **(f)** MIP-F2

Figure 9: **Heat Maps of SED error against query size and SED values for `Amazon`. Darker means higher error.**

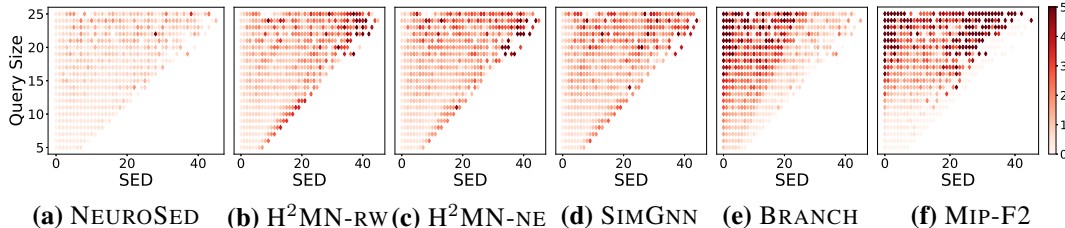

**(a)** NEUROSED **(b)** H$^2$MN-RW **(c)** H$^2$MN-NE **(d)** SIMGNN **(e)** BRANCH **(f)** MIP-F2

Figure 10: **Heat Maps of SED error against query size and SED values for `PubMed`. Darker means higher error.**

## N   VISUALIZATION AND APPLICATION OF SED

Searching for molecular fragment containment is a routine task in drug discovery (Ranu et al., 2011). Motivated by this, we show the top-5 matches to an SED query on the `AIDS` dataset produced by NEUROSED in Fig. 18. The query is a functional group (Hydrogen atoms are not represented). NEUROSED is able to extract chemical compounds that contain this molecular fragment (except for ranks 3 and 5, which contain this group with 1 edit) from around 2000 chemical compounds with varying sizes and structures. This validates the efficacy of NEUROSED at a semantic level.

## O   ALTERNATIVE QUERY DISTRIBUTIONS

In RWR, we perform fixed length random walks, where the length of a walk is the average diameter size of the queries generated through BFS during training. Next, we merge the walks to form a graph.

In RW, we perform random walks, till the diameter of the resultant graph is the same as the average diameter of the BFS sampled train graphs.

The details of the SHADOW sampler is explained in Zeng et al. (2021).

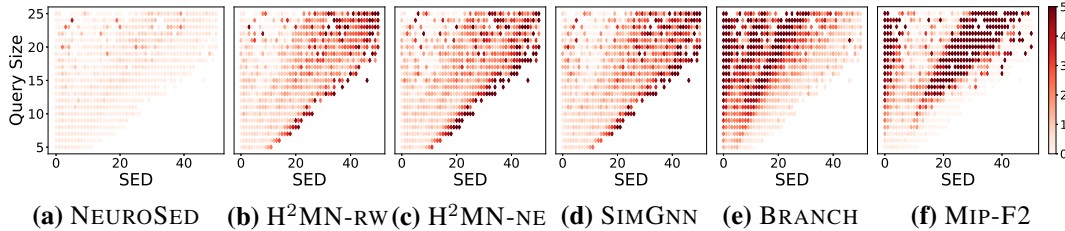

**(a)** NEUROSED **(b)** H$^2$MN-RW **(c)** H$^2$MN-NE **(d)** SIMGNN **(e)** BRANCH **(f)** MIP-F2

Figure 11: **Heat Maps of SED error against query size and SED values for `CiteSeer`. Darker means higher error.**

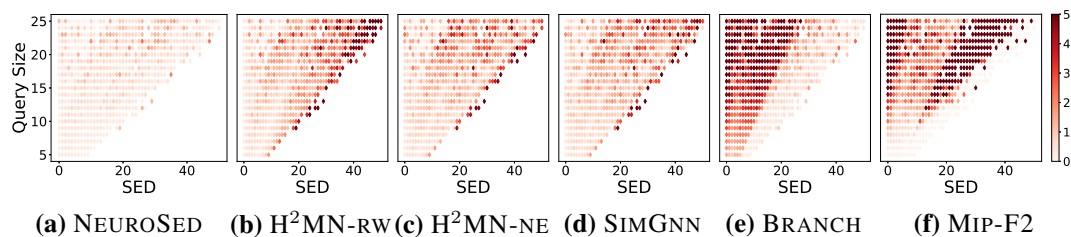

**(a)** NEUROSED **(b)** H$^2$MN-RW **(c)** H$^2$MN-NE **(d)** SIMGNN **(e)** BRANCH **(f)** MIP-F2

Figure 12: **Heat Maps of SED error against query size and SED values for `Cora_ML`. Darker means higher error.**

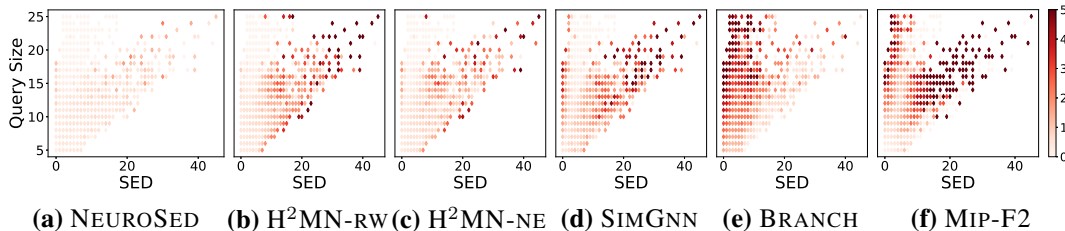

**(a)** NEUROSED **(b)** H$^2$MN-RW **(c)** H$^2$MN-NE **(d)** SIMGNN **(e)** BRANCH **(f)** MIP-F2

Figure 13: **Heat Maps of SED error against query size and SED values for `Protein`. Darker means higher error.**

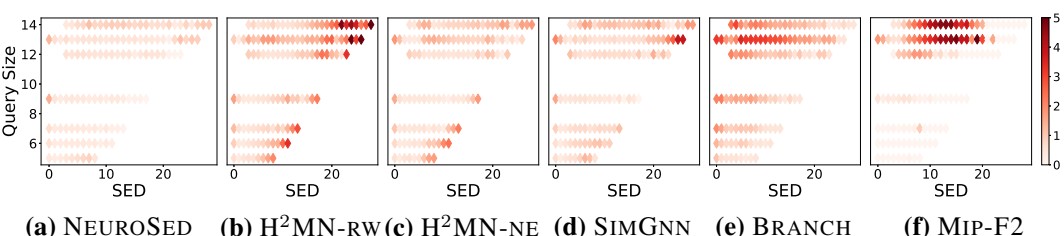

**(a)** NEUROSED **(b)** H$^2$MN-RW **(c)** H$^2$MN-NE **(d)** SIMGNN **(e)** BRANCH **(f)** MIP-F2

Figure 14: **Heat Maps of SED error against query size and SED values for `AIDS`. Darker means higher error.**

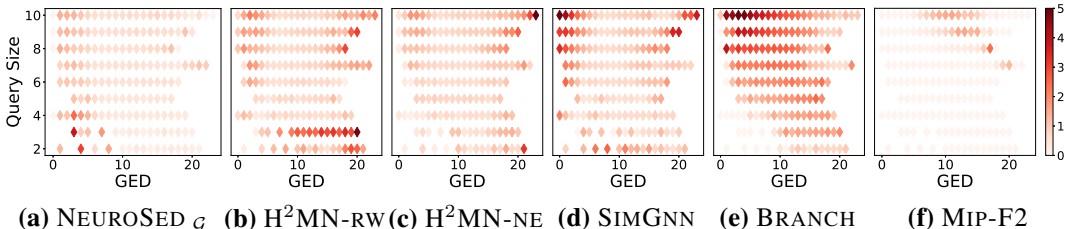

Figure 15: **Heat Maps of GED error against query size and GED values for `AIDS'`. Darker means higher error.**

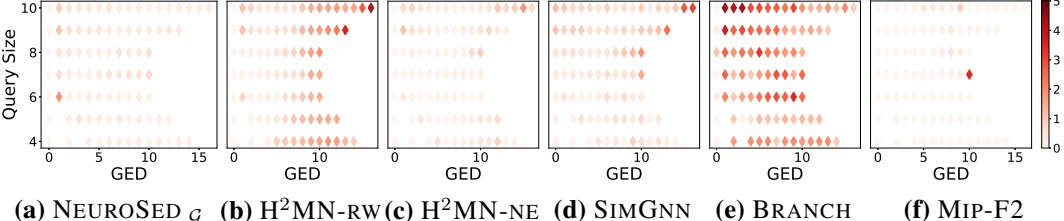

Figure 16: **Heat Maps of GED error against query size and GED values for `Linux`. Darker means higher error.**

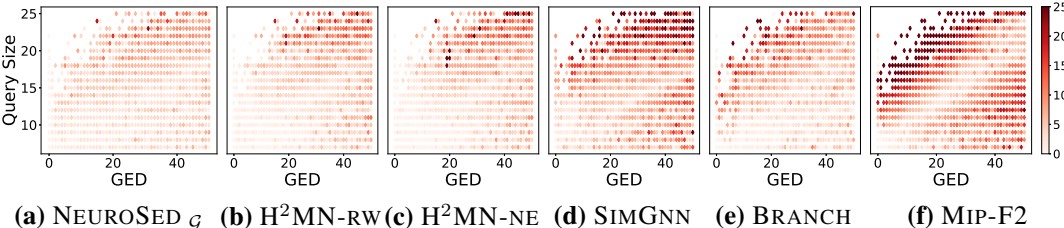

(a) NEUROSED $_\mathcal{G}$    (b) H²MN-RW (c) H²MN-NE    (d) SIMGNN    (e) BRANCH    (f) MIP-F2

Figure 17: **Heat Maps of GED error against query size and GED values for IMDB. Darker means higher error.**

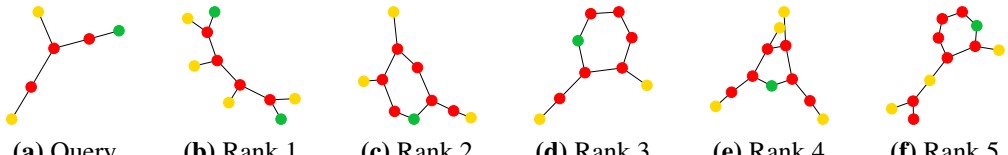

(a) Query  (b) Rank 1  (c) Rank 2  (d) Rank 3  (e) Rank 4  (f) Rank 5

Figure 18: **Visualizations of query and resulting matches produced by NEUROSED. Red, Green and Yellow colors indicate Carbon, Nitrogen and Oxygen atoms respectively. The actual and predicted SED for the target graphs are (b)** $0, 0.4$**, (c)** $0, 0.5$**, (d)** $1, 0.6$**, (e)** $0, 0.6$ **and (f)** $1, 0.6$**.**

