# OpenReview forum: "NeuroSED: Learning Subgraph Similarity via Graph Neural Networks"
_ICLR.cc/2022/Conference — ICLR 2022 Submitted_

### Official Review · Reviewer_gqTK · 2021-10-29

**Correctness:** 3
**Technical Novelty And Significance:** 3
**Empirical Novelty And Significance:** 3
**Recommendation:** 6
**Confidence:** 4

**Main Review:**

My minor concerns are:

1). Some notations are not used after definition. For example, what is π? I can only find the definition in the appendix PROOF OF LEMMA 5. This will make the reader hard to follow. Please check all notations and make sure all usage happens after a clear definition.

2). Missing some related work, such as the following one. [1] uses random walk sampling paths and model the graph as a set of paths to measure the distance between graphs.
[1] Inductive and unsupervised representation learning on graph structured objects
L Wang, B Zong, Q Ma, W Cheng, J Ni, W Yu, Y Liu, D Song, H Chen, ...
International conference on learning representations, 2020

Besides, the following paper although is cited, it is suggested to discuss the superiority of the method over the work listed below in the introduction or related work section.
[2] Rex Ying, Andrew Wang, Jiaxuan You, Chengtao Wen, Arquimedes Canedo, Jure Leskovec NEURAL SUBGRAPH MATCHING

My major concerns are the draft includes too many typos and grammar errors, I listed some. Please fix all.
 typos:
L(v) and L(e) denotes==>L(v) and L(e) denote
satisfies triangle inequality==> satisfies the triangle inequality
An edit can be addition or deletion ==>An edit can be the addition or deletion
that G1 is subgraph isomorphic==>that G1 is a subgraph isomorphic
it satisfies triangle inequality==>it satisfies a triangle inequality
high generalisation accuracy==>high generalization accuracy
denote a labelled==>denote a labeled

**Summary Of The Paper:**

The submission proposed to use GNN to encode graphs and calculate graph edit distance and subgraph edit distance based upon supervised learning with the generated graph embeddings.

**Summary Of The Review:**

In general, the idea is novel and interesting. The authors also provide proof of the properties of the proposed metric. Basically, SUBGRAPH search and matching is an important problem. The work well validated the effectiveness to use GNN for this problem. I weakly champion the acceptance.

---

> ### Author Response · Authors · 2021-11-23
> **Reviewer 4 gqTK**
>
> We thank the reviewer for the encouraging review and the critical comments that help us improve the paper further. We have uploaded an updated version incorporating the suggestions of all reviewers. All changes in the updated version are in **blue** color.
>
> 1. **Some notations are not used after definition. For example, what is π? I can only find the definition in the appendix PROOF OF LEMMA 5. This will make the reader hard to follow. Please check all notations and make sure all usage happens after a clear definition.**
>
> Answer: We introduced $\pi$ in Definition 1. To remove room for ambiguity, we have introduced a table in Appendix to summarize all frequently used notations in our work (Table 6). This table is referenced right at the start of Section 2.
>
> 2. **Missing some related work, such as the following one. [1] uses random walk sampling paths and model the graph as a set of paths to measure the distance between graphs. [1] Inductive and unsupervised representation learning on graph structured objects L Wang, B Zong, Q Ma, W Cheng, J Ni, W Yu, Y Liu, D Song, H Chen, ... International conference on learning representations, 2020**
>
> Answer: We have now referred and discussed this work in Section 1.
>
> 3. **Besides, the following paper although is cited, it is suggested to discuss the superiority of the method over the work listed below in the introduction or related work section. [2] Rex Ying, Andrew Wang, Jiaxuan You, Chengtao Wen, Arquimedes Canedo, Jure Leskovec NEURAL SUBGRAPH MATCHING**
>
> Answer:
> * We have demonstrated empirical superiority over the above mentioned paper (NeuroMatch) on top-$k$ queries for SED (See Table 3 and Fig. 7 in Appendix). We cannot compare with NeuroMatch for SED prediction accuracy (RMSE) or range queries since it returns a score that is not equivalent to SED. Thus, this score may be used for ranking, but not a substitute for SED itself.
> * In the updated version of our work, we have further enriched this comparison through a technical discussion on the two different algorithms in Section 3.3.
> * We also note that NeuroMatch cannot be applied for GED, and hence is less generic than the proposed technique.
>
> 5. **My major concerns are the draft includes too many typos and grammar errors, I listed some. Please fix all. typos: L(v) and L(e) denotes =>L(v) and L(e) denote satisfies triangle inequality => satisfies the triangle inequality An edit can be addition or deletion => An edit can be the addition or deletion that G1 is subgraph isomorphic =>that G1 is a subgraph isomorphic it satisfies triangle inequality =>it satisfies a triangle inequality high generalisation accuracy =>high generalization accuracy denote a labelled =>denote a labeled**
>
> Answer: We apologize for the typos and grammar errors. We have done a thorough proof reading and corrected all errors that came to our notice including the ones mentioned above.
>
> **Final Note:** We are encouraged by the reviewer's willingness to champion our work. Please consider raising the score if your concerns are addressed and you find this work interesting to the community or of possible practical impact.

---

### Official Review · Reviewer_Y8L2 · 2021-11-02

**Correctness:** 3
**Technical Novelty And Significance:** 2
**Empirical Novelty And Significance:** 2
**Recommendation:** 5
**Confidence:** 4

**Main Review:**

Strengths
- Efficiently computing SED is a vital problem. NeuroSED is a promising model that supports accurate and fast computation. It is a good contribution to the SED field.
- The learned embeddings hold key properties of SED, such as triangle inequality. These embeddings also support indexing strategies for fast prediction.
- Extensive experiments show how NeuroSED is a better choice than SOTA neural and non-neural methods. Sufficient ablation studies on each component of NeuroSED give evidence of why NeuroSED works.
- The paper is fairly well written.

Weaknesses
- Technique contribution is limited.
  * Although the paper argues the model is inspired by Siamese networks, it is indeed just using a shared GNN to encode both target and query graphs. In the field of combinatorial optimization, encoding graphs using GNN is straightforward and not new (Cappart et al., 2021).
  * Besides, NeuroSED is very similar to Neural Subgraph Matching (NeuroMatch, Lou et al., 2020): both models use a shared GNN to encode target and query graphs, the learning metric of NeuroSED is simply the violation penalty in NeuroMatch. NeuroMatch also studies subgraph relations in terms of subgraph embeddings as in this paper. From this regard, it seems like NeuroSED merely adopt NeuroMatch to compute SDE. Clarifications are needed.

- An advantage of neural methods over heuristic methods is the ability to efficiently predict large graphs. However, it is unclear how to extend the model to predict large graphs accurately.

  * The model is supervised, but obtaining training data for SED in practice is nontrivial. NeuroSED has to train on small target and query graph pairs with labels computed from a mixed integer programming (Lerouge et al., 2017), which is at high costs and not scalable to large graphs. As a result, training NeuroSED on large target and query graphs may be infeasible.

  * If NeuroSED trains on small graphs and predicts on large graphs, the prediction performance can be bad due to the extrapolation capacity of GNNs (Xu et al., 2021).

  * The experiments are limited to small target and query graphs with graph diameters at most 10. Showing the performance on large graphs is necessary.

- The constructed training data may have a different distribution than test data distribution in real-world applications. The training data is a collection of (target graph, query graph, label) triples, where the query graph is generated using BFS traversal on target graphs. In practice, the underlying query graph distribution can be unknown and be much different from these BFS traversal samples. Then the training distribution is different than the test distribution, making the trained model not generalized to test data.

- Pair-independent embeddings condense the target/query graph into a single vector seems to be too simple. It pushes much burden on the expressiveness of GNNs. GNNs are forced to summarize the target graph according to a large amount of query graphs. However, GNNs may not be that flexible, e.g., it is difficult for GNNs to identify simple substructures (Chen et al., 2020). It would be better to understand what features are captured in the trained GNN embeddings and ensure the pair-independent embeddings are sufficient for SED.

**Summary Of The Paper:**

The paper proposes a supervised model, NeuroSED, to compute Subgraph Edit Distance (SED) (and Graph Edit Distance (GED)). To this end, given two graphs, target and query graphs, NeuroSED uses a shared GNN to encourage embeddings for both graphs representing similar topological features. Then a dedicated learning metric w.r.t. SED is introduced to ensure the learned embeddings hold key properties of SED and the embeddings are indexable. The indexable embeddings further support fast prediction. Extensive experiments show NeuroSED significantly improves SOTA methods.

**Summary Of The Review:**

I think there is still room to improve the paper. The model seems reasonable, and the experiments show promising results under specific settings. However, the technical novelty, strategies of generating training data, and the ability to generalize the model to large graphs are not well discussed.

---

> ### Author Response · Authors · 2021-11-23
> **Reviewer 3 Y8L2**
>
> We thank the reviewer for the insightful comments. Below, we discuss how they have been addressed in the revised version. All changes in the updated version are in **blue** color.
>
> 1. **Although the paper argues the model is inspired by Siamese networks, it is indeed just using a shared GNN to encode both target and query graphs. In the field of combinatorial optimization, encoding graphs using GNN is straightforward and not new (Cappart et al., 2021).**
>
> Answer. A siamese network, by definition, means a neural network that contains one or more identical sub-networks. This is indeed the case in our architecture, where the sub-networks are GNNs. In our work, we do not claim to be the first to use siamese GNNs. However, shared weights due to siamese architecture lies at the core of our ability to preserve theoretical distance properties of the original graph space in the embedded space. We have emphasized this aspect in the updated version (see the discussion following Theorem 2 in Sec 3.2).
>
> 2. **Besides, NeuroSED is very similar to Neural Subgraph Matching (NeuroMatch, Lou et al., 2020): both models use a shared GNN to encode target and query graphs, the learning metric of NeuroSED is simply the violation penalty in NeuroMatch. NeuroMatch also studies subgraph relations in terms of subgraph embeddings as in this paper. From this regard, it seems like NeuroSED merely adopt NeuroMatch to compute SDE. Clarifications are needed.**
>
> Answer.  Both NeuroSED and NeuroMatch use siamese GNNs. The violation penalty is also present in both NeuroMatch and the SED version of NeuroSED. Nonetheless, there are important differences in (1) their objective, (2) how prediction is performed, (3) training procedure, and (4) theoretical characterization. We have summarized the differences in detail in Sec. 3.3.
>
>
> 3. **An advantage of neural methods over heuristic methods is the ability to efficiently predict large graphs. However, it is unclear how to extend the model to predict large graphs accurately.**
>
>
> Answer: We have predicted on graphs containing up to 1.66 M nodes and 7.2M edges (DBLP). This significantly extends the state of the art. Specifically:
>
> * The largest graph dataset used in any of the existing neural architectures for GED/SED prediction [H2MN-KDD 2021, GMN-ICML 2019, SIMGNN-WSDM 2019, NeuroMatch, GENN-CVPR 2021, GraphSim-AAAI-2020] is <$2500$ nodes. This is $>500$ times smaller than what we have demonstrated in our studies. Compared to the state-of-the-art neural method H2MN, NeuroSED is more than $1000$ times faster in SED and up to $20$ times faster in GED (See Sec 4.3 and Table 5a).
> * Compared to non-neural methods, NeuroSED is $100$ times faster when the graph database is given at query time and $10000$ times faster if the database is known apriori.

---

> > ### Author Response · Authors · 2021-11-23
> > **Part 2**
> >
> > 4. **The model is supervised but obtaining training data for SED in practice is nontrivial. NeuroSED has to train on small target and query graph pairs with labels computed from a mixed integer programming (Lerouge et al., 2017), which is at high costs and not scalable to large graphs. As a result, training NeuroSED on large target and query graphs may be infeasible. If NeuroSED trains on small graphs and predicts on large graphs, the prediction performance can be bad due to the extrapolation capacity of GNNs (Xu et al., 2021).**
> >
> > Answer:  In general, we agree to the above comment. Scalability to large query graphs is indeed a challenge, and as correctly pointed out, one of the root causes is the generation of training data. The scalability issue applies to *all* neural approaches for graph distance prediction since both GED and SED are NP-complete. However, there are certain caveats.
> > * **Generalizability to large target graphs:** If the query graphs are not too large (<50 nodes and 150 edges), it is not hard to generalize to large target graphs. Particularly, the target graph can be decomposed into small node neighborhoods and the query may be scanned with each of these neighborhoods. As long as the neighborhood is sufficiently large (See Theorem 3 in Appendix for the specifics), this approach returns the optimal answer. This property lies at the core of generalizing well to large target graphs.
> > * **Generalizability to large query graphs:** This is a harder problem to solve.
> >     * **New experiments:** We have added new experiment to further evaluate scalability and do not observe any sharp deterioration till query sizes containing at least 50 nodes (See Appendix M of the updated version). Existing works are limited to queries of $\approx 20$ nodes. Furthermore, we do scale better with query graph sizes in terms of quality than the baselines (See heatmaps in Appendix L). Thus, we do push the boundary.
> >     * **Ability to generalize:** The ability to train on small queries and test on larger queries remains an unsolved problem. To further study this aspect, we attempted this and the results indicate weak generalization (Appendix M). We now mention this as an explicit limitation of our work in Conclusion with pointer to the Appendix on the empirical data (this limitation also applies to other neural approaches). In NeuroMatch, curriculum learning is used to circumvent this problem. In our experience, curriculum learning did not help.
> >     * **Small queries are common:** While generalizability to large queries is a desirable property, there are plenty of domains where small query graphs are common (<50 nodes). For example, the size of $99\%$ graph queries on the WikiData knowledge graph is less than $11$ [1]. Chemical compounds typically contain less than 26 nodes [2]. Finally, in the well known graph benchmark dataset, TUDataset, 14 out of 16 datasets have graph sizes below $50$ [3]. Hence, NeuroSED promises to be of help in these application domains.
> >
> > [1] An analytical study of large SPARQL query logs, Angela Bonifati, Wim Martens, Thomas Timm, Proceedings of the VLDB Endowment, Volume 11 Issue 2, October 2017 pp 149–161, https://doi.org/10.14778/3167892.3167895.
> >
> > [2] https://zinc.docking.org/
> >
> > [3] TUDataset: A collection of benchmark datasets for learning with graphs, Christopher Morris, Nils M. Kriege, Franka Bause, Kristian Kersting, Petra Mutzel, Marion Neumann, ICML 2020 workshop "Graph Representation Learning and Beyond"
> >
> > 5. **The experiments are limited to small target and query graphs with graph diameters at most 10. Showing the performance on large graphs is necessary.**
> >
> > Answer. Please refer to our answers to Comments 3 and 4 above. We have shown scalability to target graphs containing millions of nodes. In contrast, the largest dataset used in any of the previous or baseline neural approaches for GED or SED is 2500, with a large majority limited to graphs less than 50 nodes.

---

> > > ### Author Response · Authors · 2021-11-23
> > > **Part 3**
> > >
> > > 6. **The constructed training data may have a different distribution than test data distribution in real-world applications. The training data is a collection of (target graph, query graph, label) triples, where the query graph is generated using BFS traversal on target graphs. In practice, the underlying query graph distribution can be unknown and be much different from these BFS traversal samples. Then the training distribution is different than the test distribution, making the trained model not generalized to test data.**
> > >
> > > Answer. The proposed technique generalizes well to unseen distributions. This is evident from the following experiments:
> > >
> > > * All queries for the AIDS dataset are real functional groups (mentioned in "Queries" para of Sec 4.1 ). The model is trained on BFA sampled queries. We observe the errors are consistently low (Table 2a and 3a, Fig 3e). This indicates good generalization to unseen distributions.
> > >
> > > * We have added new experiments, where the model is trained on BFS, but evaluated on the three unseen distributions of Random Walks, Random Walks with Restarts and shaDow [4]. As evident from Sec 4.5, the errors remain low.
> > >
> > > [4] "Decoupling the Depth and Scope of Graph Neural Networks", Hanqing Zeng, Muhan Zhang, Yinglong Xia, Ajitesh Srivastava, Andrey Malevich, Rajgopal Kannan, Viktor Prasanna, Long Jin, Ren Chen, NeurIPS 21.
> > >
> > > 7. **Pair-independent embeddings condense the target/query graph into a single vector seems to be too simple. It pushes much burden on the expressiveness of GNNs. GNNs are forced to summarize the target graph according to a large amount of query graphs. However, GNNs may not be that flexible, e.g., it is difficult for GNNs to identify simple substructures (Chen et al., 2020). It would be better to understand what features are captured in the trained GNN embeddings and ensure the pair-independent embeddings are sufficient for SED.**
> > >
> > > Answer:
> > >
> > > * First, we highlight the fact that other neural baselines also compute pair-independent embeddings. However, the similarity between embeddings is computed in a pair-dependent manner through a neural network. In contrast, we use a non-neural and indexable function directly on the embeddings. That's why in our paper we say
> > > >"There is little scope for pre-computation in existing approaches (either neural or non-neural), as the major **computations** are pair-dependent, i.e., both $G_1$ and $G_2$ need to be known."
> > >
> > > Thus, pair-independent embeddings are also used in other neural approaches.
> > >
> > > * Indeed summarizing a large target graph into a single embedding is hard. Thus, we break it up into neighborhoods (Recall Theorem 3 in Appendix) and essentially represent a large target graph containing $n$ nodes with $n$ embeddings. We also see deterioration in quality with increase in node set size. This indicates that the next step in this line of work is to learn embeddings that scale better with graph sizes (Nonetheless, even in this aspect NeuroSED is better as discussed above).
> > >
> > > * Regarding interpretability, it would indeed be an interesting study to understand the features being captured by the various embedding architectures (including ours). However, we feel this would require an independent study of its own and is beyond the scope of the current work.
> > >
> > >
> > > **Final Note:** Please consider raising the score if your concerns are addressed and you find this work interesting to the community or of possible practical impact. To reiterate, the key contributions are:
> > > * **Impact on GED:** In addition to SED, learning GED has seen significant interest in the community [H2MN-KDD 2021, GMN-ICML 2019, SIMGNN-WSDM 2019, NeuroMatch, GENN-CVPR 2021, GraphSim-AAAI-2020], and the proposed technique comprehensively improves upon the state of the art (H2MN, KDD 2021) both on quality and querying time.
> > >
> > > * **Scalability with Query Size:** We agree that generalization to large query sizes is a limitation of our work. We explicitly acknowledge this in our Conclusions. However, this limitation applies to all baselines as well. Furthermore, the proposed work generalizes better than the baselines.
> > >
> > > * **Similarity with NeuroMatch:** There are similarities with NeuroMatch. However, the characterization is richer in our work resulting in superior generalizability to GED and SED, as well as better empirical performance. We now crisply discuss this aspect.
> > >
> > > * **Efficiency:** In terms of raw efficiency, the proposed approach is orders of magnitude faster and scales to million-sized target graphs.

---

### Official Review · Reviewer_ECCJ · 2021-11-02

**Correctness:** 4
**Technical Novelty And Significance:** 3
**Empirical Novelty And Significance:** 4
**Recommendation:** 6
**Confidence:** 3

**Main Review:**

Strengths:
1. Simple yet effective approach for estimating the SED from two graph embeddings.
2. The adopted score function Eq. (6) can preserve SED's properties of nonnegativity, subgraph identity, and triangle inequality, which is theoretically proved.
3. Solid experiments show state-of-the-art performance and orders of reduction in computation time.
4. The writing is clear. The paper is easy to follow and demonstrates the key insights very well.

Weaknesses:
1. Equation (6) seems the key technical innovation of the proposed model, which seems a bit simple and appears to be the same one as in Neural Subgraph Matching [1]. Although theoretical analysis is given to show it satisfies three properties in Lemma 1. However, it should not be the only one satisfying the three properties. Then why is Eq. (6) exclusively better? Is it possible to improve (6) further? A discussion of [1] should also be included.
2. The title of Table 4 is too close to Table 5, making it look like the title of Table 5.

[1] Lou, Zhaoyu, et al. "Neural Subgraph Matching." arXiv preprint arXiv:2007.03092 (2020).

**Summary Of The Paper:**

This paper presents a simple and effective model for subgraph similarity search. Two GNNs with shared weights are applied first to get the graph embeddings of the query and target graphs. Then, instead of using an MLP over the concatenated graph embeddings to regress the subgraph edit distance (SED) score, the paper proposes to use the l2 norm of the positive portion of the difference between two embeddings. This inductive bias is proved to preserve SED's triangle inequality property, and shows better accuracy and generalization power for large graphs. Experiments show the proposed model outperforms state-of-the-art neural approaches in both graph similarity search and subgraph similarity search tasks.

**Summary Of The Review:**

Considering the strong empirical performance and the theoretical guarantee, I recommend an acceptance. My only concern is the possible limited technical innovation in Eq. (6), which desires more in-depth analysis and discussion. Besides, it already appears in previous papers and a discussion is lacking.

---

> ### Author Response · Authors · 2021-11-23
> **Reviewer 2 (ECC)**
>
> We thank the reviewer for the constructive feedback. We have updated our paper with the following changes to better justify the design choices. All changes in the updated version are in **blue** color.
>
> 1. **Equation (6) seems the key technical innovation of the proposed model, which seems a bit simple and appears to be the same one as in Neural Subgraph Matching [1]. Although theoretical analysis is given to show it satisfies three properties in Lemma 1. However, it should not be the only one satisfying the three properties. Then why is Eq. (6) exclusively better? Is it possible to improve (6) further? A discussion of [1] should also be included.
> [1] Lou, Zhaoyu, et al. "Neural Subgraph Matching." arXiv preprint arXiv:2007.03092 (2020).**
>
> Answer:
>
> * **Choice of Eq. 6:** We have significantly expanded our discussion on the choice of Eq. 6 (See Sec 3.2 with details in App E). Specifically, we derive why the proposed prediction functions are a natural choice, and what are the core properties that any alternative function must satisfy. Our analysis shows that under some mild technical assumptions which guarantee technical feasibility, any embedding space prediction function which models SED (or GED) has a form which coincides with Eq. 6 (or Eq. 7) modulo the choice of norm. Any monotonic norm suffices, and the exact choice can be seen as a hyperparameter. L2 norm is a natural choice which also gives good empirical results.
>
> * **Comparison to NeuroMatch[1]:** We have demonstrated empirical superiority over the above-mentioned paper (NeuroMatch) on top-$k$ queries for SED (See Table 3 and Fig. 7 in Appendix). We cannot compare with NeuroMatch for SED prediction accuracy (RMSE) or range queries since it returns a score that is not equivalent to SED. Thus, this score may be used for ranking, but not a substitute for SED itself. In the updated version of our work, we have further enriched this comparison through a technical discussion on the two different algorithms in Section 3.2. Finally, while we generalize to both GED and SED, NeuroMatch cannot be used for GED.
>
>
> 2. **The title of Table 4 is too close to Table 5, making it look like the title of Table 5.**
>
> Answer: We have corrected this presentation issue in the updated version.

---

> > ### Comment · Reviewer_ECCJ · 2021-11-30
> > **Response to author**
> >
> > Thanks the authors for providing the response. I feel Eq. 6 should be discussed more thoroughly and [1] should be given more attention in the revised paper. The main derived loss function Equation (6) is essentially the same one used in [1]. Therefore the technical novelty may be limited, and more discussion on the difference and relation to [1] should be paid.

---

> > > ### Author Response · Authors · 2021-11-30
> > > **Similarity to NeuroMatch's loss function.**
> > >
> > > We acknowledge this feedback. Indeed, the loss functions are similar for NeuroSED and NeuroMatch and we do not claim novelty in this aspect. Our contribution lies in characterizing the loss function, and why it is a natural choice for the proposed task (App E) along with other norms that may also suffice.
> > >
> > > Furthermore, Eq. 6 is specific to only SED. However, our contribution lies not only on SED, but also on GED. Learning GED has seen significant interest in the community [H2MN-KDD 2021, GMN-ICML 2019, SIMGNN-WSDM 2019, GENN-CVPR 2021, GraphSim-AAAI-2020], and the proposed technique comprehensively improves upon the state of the art (H2MN, KDD 2021) both on quality and querying time.
> > >
> > > We will emphasize this further in the revised version.

---

### Official Review · Reviewer_L1VP · 2021-11-03

**Correctness:** 3
**Technical Novelty And Significance:** 2
**Empirical Novelty And Significance:** 3
**Recommendation:** 5
**Confidence:** 4

**Main Review:**

Strong points:

- The paper is well presented and organized. It is easy to follow with adequate examples.

- The ideas are intuitive and technically sound, although alternative or even better solutions could exist.

- There is a good balance of empirical and theoretical results.

Weak points:

- The paper emphasizes a lot on the ability to preserve theoretical properties. But these theoretical properties are direct consequences of the prediction function, and has nothing to do with the rest of the model (e.g. the siamese network, the GNN layers). In theory, any neural network architecture, followed by the same prediction function, would preserve these theoretical properties.

- The prediction function is manually crafted. Why not make it learnable? In theory, an MLP can universally approximate any continuous function in a specified input range--- which could automatically discover the right prediction function, including the proposed handcrafted function. To improve learning, as a prior, some regularization can be added to guide the MLP toward the theoretical properties.

- Some important neural baselines are missing for SED.

[a] Neural Subgraph Isomorphism Counting. KDD2020.

[b] Subgraph Neural Networks. NeurIPS 2020.

Both are based on graph neural networks. Although they are not for SED, they are for subgraph isomorphism counting, the difference only lies in the loss function. If they are replaced with the same loss on SED, they can be reasonable baselines.

- The pair-independent advantage is not unique to this method (see [a]). Moreover, it may not be that useful. First, it is only applied at inference time, which is typically fast for neural networks. Second, this only applies to cases where query/graph repeatedly appear in different combinations from a fixed pool. In cases where each incoming query/graph is new, the model has to compute their embeddings on the fly anyway.

Minor issue:

- F are sometimes used instead of \mathcal{F}.

--- update after rebuttal ---
I see some valid points in the rebuttal, hence upgrading the score.

**Summary Of The Paper:**

The paper addresses the problem of graph/subgraph similarity search in terms of the edit distance, termed GED/SED, respectively. Unfortunately, both GED and SED are NP hard, with exponential search space, making the cost prohibitive on even moderately large graphs/queries. Instead, the authors apply a neural network based approach, by designing a siamese graph neural network called NEUROSED. In particular, NEUROSED is able to preserve theoretical properties of  SED/GED, such as triangle inequality, through a purposely designed prediction function. Experiments are conducted on real graph datasets for both SED and GED, which evaluates the effectiveness and efficiency of GED/SED.

**Summary Of The Review:**

Based on the weaknesses stated above, the handcrafted prediction function is the key but there may be better alternatives. Additionally, other subgraph-based GNN can be easily ported over to solve the SED problem.

---

> ### Author Response · Authors · 2021-11-23
> **Reviewer 1 L1VP**
>
> Thank you for your constructive comments on our work. Below we address the issues raised.  All changes in the revised draft are in **blue** color.
>
> **1. The paper emphasizes a lot on the ability to preserve theoretical properties. But these theoretical properties are direct consequences of the prediction function, and has nothing to do with the rest of the model (e.g. the siamese network, the GNN layers). In theory, any neural network architecture, followed by the same prediction function, would preserve these theoretical properties.**
>
> Answer: It would be incorrect to say that any neural architecture followed by the proposed prediction function would preserve these theoretical properties. Specifically, the neural architecture *must* ensure the following:
> * **Pair-indepedent representation:** Since all of the existing neural architectures for GED and SED are pair-dependent, if the final distance prediction is done using our proposed function but using a prior architecture, the theoretical properties would *not* be preserved.
> * **Shared weights in the siamese architecture:** Unless the representations of the query and target graphs share the same parameters, they would violate the shown properties.
>
> We emphasize these more clearly in the discussion for Theorem 2 in Section 3.2 of the updated version.
>
> **2. The prediction function is manually crafted. Why not make it learnable? In theory, an MLP can universally approximate any continuous function in a specified input range--- which could automatically discover the right prediction function, including the proposed handcrafted function. To improve learning, as a prior, some regularization can be added to guide the MLP toward the theoretical properties.**
>
> Answer: We have compared with MLP (See Section 4.4 and Fig. 4) and shown that the inductive bias introduced through our prediction function yields better results, particularly at low volumes of training data. Our prediction functions yield better results since it allows better generalizability from low volumes of training data. This is a desirable property in the context of graph distance learning since optimal computation of both GED and SED are NP-hard.
>
> Indeed, MLPs are universal approximators and, in theory, should be able to learn the proposed function. However, the volume of training data required to learn the function can be arbitrarily large. The inductive bias introduced through our function overcomes this limitation.
>
> In addition, the prediction function guarantees preservation of SED and GED properties from the original graph space, which is not possible through an MLP. While regularization may help, it can never guarantee properties such as the distance in the embedded space being metric (GED) and satisfying triangle inequality (SED). Losing these properties hampers higher orders tasks (See Sec 1, Page 2) and the ability to expedite search through indexing.
>
> Nonetheless, we have further expanded our discussion on the prediction function by characterizing why it is a natural choice for the proposed problem, and if any alternatives exist, what are the basic properties that the alternative function must satisfy (See Sec 3.2 with details in Appendix E).

---

> > ### Author Response · Authors · 2021-11-24
> > **Part 2**
> >
> > **3. Some important neural baselines are missing for SED.**
> >
> > * **[a] Neural Subgraph Isomorphism Counting. KDD2020.**
> > * **[b] Subgraph Neural Networks. NeurIPS 2020.**
> >
> > **Both are based on graph neural networks. Although they are not for SED, they are for subgraph isomorphism counting, the difference only lies in the loss function. If they are replaced with the same loss on SED, they can be reasonable baselines.**
> >
> > Answer: First, we note that paper [b] (Subgraph Neural Networks. NeurIPS 2020) models aspects of subgraphs that are orthogonal to the requirements of predicting SED. Specifically, in [b], the goal is to represent subgraphs within the context of a larger graph. Hence, representation of a subgraph is dependent on the messages it receives from other *anchor subgraphs* of the same large graph. This approach makes it unsuitable for the proposed problem since we will not even be able to embed the query graph as it is not a subgraph of a larger graph. In our problem, the goal is to embed whole graphs such that by comparing the representations of the whole graphs we can predict GED/SED.
> >
> > Regarding [a], its scope is more limited compared to NeuroSED since NeuroSED models both SED and GED (where we improve upon the state of the art), whereas [a] is focused on subgraph level computations. Nonetheless, we agree that it could be considered as a baseline for SED by changing the loss function. Unfortunately, due to some implementation hurdles (data format in [a] is different to ours), we do not have the results yet. We are working on this and hope to present it before 29 November.
> >
> > **UPDATE (25th November):** We have added results for SED in the next comment. NeuroSED is consistently better in quality and $20$ times faster on average than [a].
> >
> > **4. The pair-independent advantage is not unique to this method (see [a]). Moreover, it may not be that useful. First, it is only applied at inference time, which is typically fast for neural networks. Second, this only applies to cases where query/graph repeatedly appear in different combinations from a fixed pool. In cases where each incoming query/graph is new, the model has to compute their embeddings on the fly anyway.**
> >
> > Answer. We respectfully disagree with the above comment. Let us elaborate.
> >
> > * **"The pair-independent advantage is not unique to this method (see [a])."**: Our paper claims to be the only pair-independent method for neural graph distance computations. [a] is not for graph (or subgraph) distance computation. Second, the computation in [a] is not pair-independent either (See Fig. 4 and Sec 3.3 in [a]). More specifically, after the initial computations of query and target graph representations, they simultaneously are passed through another gated recurrent network that is **pair-dependent**. In our case, the entire neural model is pair-independent, and the distance is a simple (non-neural) L2 distance over the embeddings (an additional ReLU filter in SED).
> >
> > * **"it is only applied at inference time, which is typically fast for neural networks."**: Our scalability experiments (Table 5a) clearly show that NeuroSED is 1000 times faster than H2MN, which is the fastest and most recent among existing neural approaches for GED/SED. In addition, even a CPU-version of NeuroSED is 50 times faster than the GPU version of H2MN.
> >
> > * **"this only applies to cases where query/graph repeatedly appear in different combinations from a fixed pool."**: The query graph does not need to appear repeatedly from a fixed pool. We embed the query at inference time. Pre-embedding is performed only on the database graph(s). This assumption of a known database is common in real life and is the core reason on why index structures exist. Practical examples of such databases include the ZINC chemical compound repository, databases of protein structures, knowledge graphs like Freebase, YAGO.
> >
> > **5. Minor issue: F are sometimes used instead of $\mathcal{F}$.**
> >
> > Thank you for pointing this out. We have now corrected this.
> >
> > **Final note:**
> > * Please consider raising the score if your concerns are addressed and you find this work interesting to the community or of possible practical impact.
> > * We also request the reviewer to consider the fact that our contribution lies not only on SED, but also on GED. Learning GED has seen significant interest in the community [H2MN-KDD 2021, GMN-ICML 2019, SIMGNN-WSDM 2019, NeuroMatch, GENN-CVPR 2021, GraphSim-AAAI-2020], and the proposed technique comprehensively improves upon the state of the art (H2MN, KDD 2021) both on quality and querying time.

---

> > > ### Author Response · Authors · 2021-11-26
> > > **Results for comparison with [a] Neural Subgraph Isomorphism Counting. KDD2020.**
> > >
> > > The table below presents the results on SED for NeuroSED, [a] (let us call **NSC**) and H2MN. As visible, NeuroSED is consistently superior. In addition, NeuroSED is also $\approx 20$ times faster on average. NSC is inferior to H2MN in 3 out of 5 datasets. This result further substantiates the claim of NeuroSED being the state-of-the-art neural algorithm for GED and SED prediction.
> > >
> > > **RMSE:**
> > >
> > > | Dataset      | AIDS | Protein | Amazon | Citeseer | Cora
> > > | ----------- | ------|-------- |--------|-----------|-----
> > > | NeuroSED      | **0.51**      |**0.52** |**0.49** | **0.52** | **0.63**
> > > | NSC   |    0.56     | 0.66|2.14 | 1.66 | 1.66
> > > | H2MN-RW | 0.75| 0.94| 1.29 | 1.5 |1.45
> > > |H2MN-NE | 0.66| 0.75| 0.97| 1.83 | 1.23
> > >
> > > **Running Time** per 10k queries in seconds. NeuroSED times are unindexed and all embeddings are done at query time. If indexed, NeuroSED will be orders of magnitude faster.
> > >
> > >
> > >
> > > | Dataset      | AIDS | Protein | Amazon| Citeseer | Cora
> > > | ----------- | ------|-------- |-      |-----------|-----
> > > | NeuroSED      | **0.84**       |**0.86** |**1.46**| **1.28** | **1.25**
> > > | NSC   |    4     | 21|21| 24.78 | 24.04
> > > | H2MN-RW | 9.63| 19.33| 23.2| 27.51 |29.04
> > > |H2MN-NE | 15.76| 28.99| 40.34| 54.46 | 70.59
> > >
> > >
> > > **Note:** We would also be happy to share the results on K-NN and range queries for NSC if you wish to see.

---

### Decision · Program_Chairs · 2022-01-20

**Decision:**

Reject

**Comment:**

The paper proposes a new method for subgraph similarity search by learning embeddings via a GNN-based approach to reflect the edit distance between subgraphs. Reviewers highlighted that the paper proposes an intuitive and promising approach to an interesting problem and provides a good balance between theoretical and empirical results. However, reviewers raised also concerns regarding the significance of technical contributions, limited analysis (e.g, performance on large-scale graphs, baselines, evaluation) and comparison to related work. After author response and discussion, reviewers did not come to a full agreement with two reviewers indicating weak acceptance and two reviewers indicating (weak) reject. Taking rebuttal and discussion into account, I agree with the viewpoint that the paper is not yet ready for acceptance at ICLR as it would require an additional revision to fully address the raised concerns. However, I encourage the authors to revise and resubmit their manuscript based on the feedback from this reviewing round.